# A Devonian Fish Tale: A New Method of Body Length Estimation Suggests Much Smaller Sizes for *Dunkleosteus terrelli* (Placodermi: Arthrodira)

Russell K. Engelman

Department of Biology, Case Western Reserve University, Cleveland, OH 44106, USA; neovenatoridae@gmail.com

**Abstract:** *Dunkleosteus terrelli*, an arthrodire placoderm, is one of the most widely recognized fossil vertebrates due to its large size and status as one of the earliest vertebrate apex predators. However, the exact size of this taxon is unclear due to its head and thoracic armor being the only elements of its body regularly preserved in the fossil record. Lengths of 5–10 m are commonly cited, but these estimates are not based on rigorous statistical analysis. Here, I estimate the body size of *D. terrelli* using a new metric, orbit-opercular length, and a large dataset of arthrodires and extant fishes (3169 observations, 972 species). Orbit-opercular length strongly correlates with total length in fishes ($r^2 = 0.947$, $PE_{cf} = 17.55\%$), and accurately predicts body size in arthrodires known from complete remains. Applying this method to *Dunkleosteus terrelli* results in much smaller sizes than previous studies: 3.4 m for typical adults (CMNH 5768) with the largest known individuals (CMNH 5936) reaching ~4.1 m. Arthrodires have a short, deep, and cylindrical body plan, distinctly different from either actinopterygians or elasmobranchs. Large arthrodires (*Dunkleosteus*, *Titanichthys*) were much smaller than previously thought and vertebrates likely did not reach sizes of 5 m or greater until the Carboniferous.

**Keywords:** Devonian; Paleozoic; Placodermi; body shape; size estimation; ichthyology; axial elongation; vertebrate size; log-transformation





## 1. Introduction

*Dunkleosteus terrelli* is a large arthrodire "placoderm" (hereafter without quotes, see [1–4], but also [5]), best known from the latest Devonian (late Famennian) Cleveland Shale of Ohio, USA. This taxon is one of the most recognizable prehistoric organisms and is by far one of the most widely known Paleozoic vertebrates, only comparable to the Permian stem-mammal *Dimetrodon limbatus* in this respect. *Dunkleosteus'* popularity largely stems from its unique morphology, which includes an extensive dermal skeleton, blade-like jaws, and large size. These features, as well as its great geologic age, result in this taxon often being considered "one of the first vertebrate superpredators" [6–9]. However, in spite of its prominence in paleo pop culture, relatively little is known about *Dunkleosteus* as an actual animal. Although some studies have been conducted on the paleobiology of *Dunkleosteus* (e.g., [8–12]), even very basic questions about this taxon such as "how large did it grow" or "what did it look like" remain unanswered.

This uncertainty largely stems from the unusual anatomy of *Dunkleosteus* and other arthrodires compared to most vertebrates. In contrast to most living vertebrates, which either have skeletons that are almost entirely cartilaginous (sharks, lampreys) or almost entirely ossified (bony fishes), arthrodires combine an ossified head and thoracic armor with a mostly cartilaginous post-thoracic skeleton (which includes the caudal region and major fins). Thus, the head and thoracic armor of arthrodires are frequently preserved in the fossil record but the rest of the body is typically lost during fossilization.

Post-thoracic remains are only known for a handful of arthrodire taxa. The best known of these is *Coccosteus cuspidatus*, a freshwater/brackish [13,14] coccosteomorph represented

by several complete specimens [15]. Extensive postcranial material and/or body outlines are also known for the non-eubrachythoracid arthrodires *Holonema westolii* [16], *Africanaspis doryssa*, and *A. edmountaini* [17]; the coccosteomorphs *Millerosteus minor*, *Dickosteus threiplandi*, *Watsonosteus fletti* [18], *Plourdosteus canadensis* [19], and *Incisoscutum ritchei* [20]; the aspinothoracidan *Amazichthys trinajsticae* [21]; and *Rhachiosteus pterygiatus*, a eubrachythoracid of uncertain phylogenetic position (see [22–24]). Limited post-thoracic material has been reported for several other arthrodires, including *Paramylostoma arcualis* [25], *Heintzichthys gouldii* [26,27], and *Dunkleosteus terrelli* itself [28,29]. However, the post-thoracic remains associated with *D. terrelli* are not extensive enough to make accurate inferences about the overall size or shape of the animal.

This lack of information about body size makes it difficult to reconstruct the paleobiology of *Dunkleosteus* and other large arthrodires, as well as the paleoenvironments in which they lived. Body size influences nearly every aspect of an organism's biology [30], from life history patterns to predator–prey relationships to something as simple as scaling morphological variables to compare species of different sizes. In the case of *Dunkleosteus* alone, body size has been considered a relevant variable in determining likely life habits [11], scaling relative bite force [31], inferring caudal fin shape [12], and examining broader patterns of vertebrate size evolution across the middle Paleozoic [32]. Indeed, this issue was even mentioned directly by Carr [11], who used body mass estimates to demonstrate *Dunkleosteus* was likely an active swimmer because it was too heavy to rest on the fine-grained seafloor in its paleoenvironment. However, Carr [11] also noted in this analysis these mass estimates were very approximate due (in part) to the lack of well-constrained size estimates for *Dunkleosteus*.

The body size of large, late Devonian arthrodires like *Dunkleosteus* has typically been estimated based on the dimensions of much smaller, distantly related arthrodires such as *Coccosteus* (but see "Body Size of *Dunkleosteus*" below). However, not only are the body proportions of *Dunkleosteus* likely very different from these arthrodires due to differences in ecology (pelagic versus mostly demersal) [12,33], but estimating the size of *Dunkleosteus* using smaller arthrodires requires a significant degree of extrapolation. The largest arthrodire for which complete body fossils are known, *Amazichthys trinajsticae*, is ≤1 m in length [21], with other complete arthrodires generally measuring between 20–60 cm (see Supplementary File S3: Table S8). This is significantly smaller than even the most conservative size estimates for *Dunkleosteus*. Solely relying on these smaller arthrodires increases the risk of extrapolation error [30], which has been known to cause inaccurate size estimates in other groups of extinct organisms [34–36]. The most recent attempt to estimate the size of *Dunkleosteus* attempted to solve these problems by extending a proxy known to reliably predict size in large sharks (upper jaw perimeter) to arthrodires [12]. However, further investigation has found these estimates are not reliable as they fail to control for anatomical differences between arthrodires and elasmobranchs (specifically, arthrodires having much larger mouths relative to body size; [33]).

*Introducing Orbit-Opercular Length*

Thus, it is clear a new method is needed to estimate size in *Dunkleosteus* and other arthrodires. Specifically, this method must accurately estimate length across fishes in general (e.g., lampreys, chondrichthyans, and bony fishes), be measurable in arthrodire fossils, and provide accurate length estimates for arthrodires known from complete remains. One such potential proxy is orbit-opercular length (hereafter OOL). This is the length from the anterior margin of the orbit to the posterior margin of the head (Figure 1). There are several biological reasons to believe OOL and total length would be highly correlated in fishes. For one, OOL encompasses two key anatomical regions: the neurocranium (including the brain and orbits) and the gill chamber. Both regions are highly important for survival in most fishes and thus their size is likely to be strongly constrained.

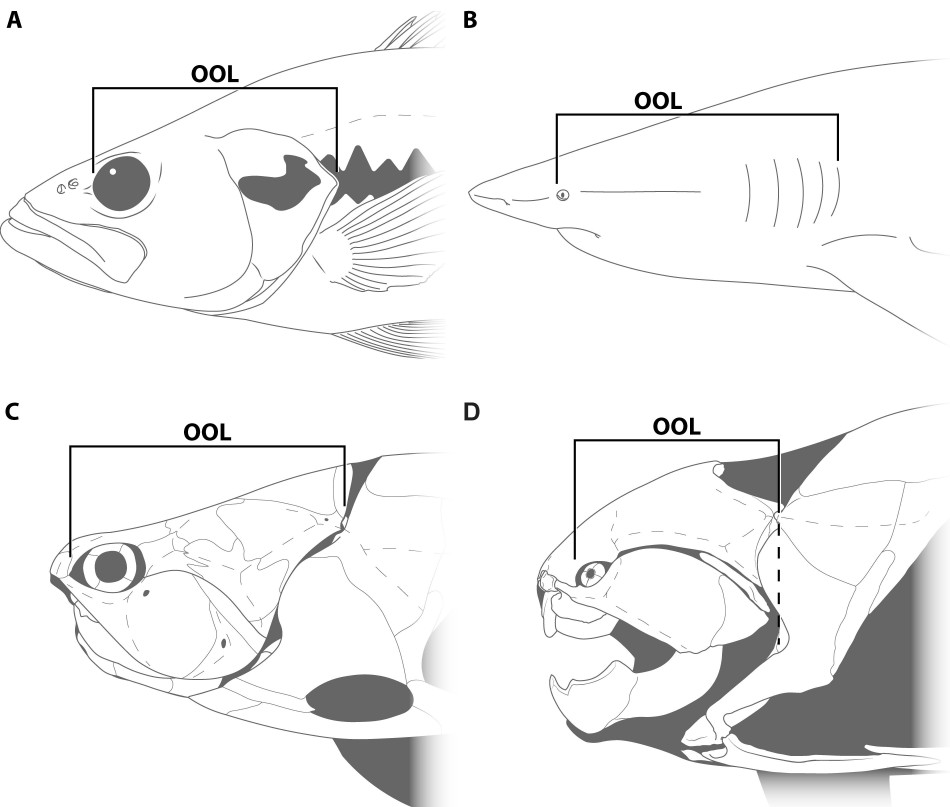

**Figure 1.** Heads of (**A**), an osteichthyan (*Micropterus dolomieu*, CMNH Teaching Collection); (**B**), a chondrichthyan (*Carcharhinus obscurus*; modeled after specimen in [37]); (**C**), a coccosteomorph arthrodire (*Coccosteus cuspidatus*, after [15]); and (**D**), *Dunkleosteus terrelli* (CMNH 6090), showing how orbit-opercular length was measured accounting for different arrangements of the gill skeleton in different lineages. Dotted line in (**C**,**D**) shows how the cranio–thoracic joint in arthrodires exceeds (in **C**) or is equivalent to (in **D**) the level of the posterior gill margin.

Although there is some interspecific variation in brain or gill size among fishes [38,39], the eyes, brain, and gills are limited in how large or small they can be relative to body size before they negatively impact fitness. If these organs are too small, they may not be able to meet functional demands (i.e., oxygen uptake for gills), whereas if they are too large they can impose unnecessary metabolic or functional costs (e.g., a gill chamber that is much larger than the fish needs for respiration). Organ sizes in fishes tend to be highly optimized in order to maintain a hydrodynamic shape [39], which further suggests these proportions should be very constrained. By contrast, the remaining proportion of the head, the snout or rostral length, is influenced by feeding habits or ecology and is expected to show more variation between species. Similar results have been observed in studies of carnivoran mammals [40] and crocodile-line archosaurs and their relatives [41,42], where neurocranial size (measured in a similar manner to OOL here) was considered more robust to interspecific differences in head proportions, due to ecologically-driven variation in snout length.

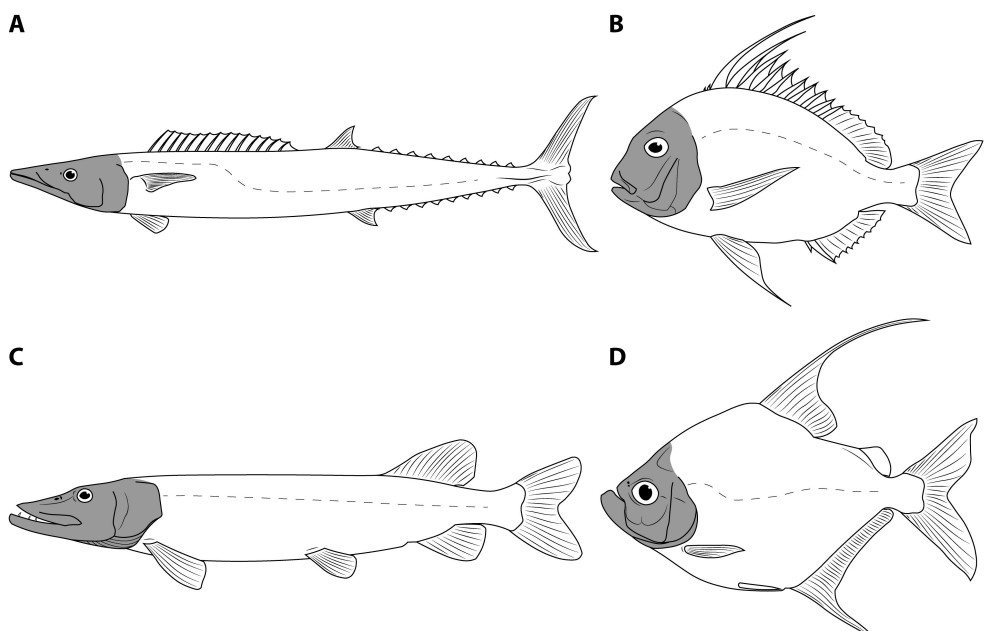

**Figure 2.** Elongate-bodied (**A**,**C**), and compressiform (**B**,**D**) acanthopterygian (**A**,**B**) and non-acanthopterygian (**C**,**D**) fishes, showing how short fishes have short heads and long fishes have long heads. (**A**), *Acanthocybium solandrei* (Scombridae), drawn from FSBC 6267. (**B**), *Catoprion abscondidus* (Serrasalmidae), modified from Bonani Mateussi et al. [43]. (**C**), *Esox lucius* (Esocidae), modified from Casselman et al. [44]. (**D**), *Argyrops spinifer* (Sparidae), modified from Randall [45]. Drawings by Russell Engelman.

Some of these theoretical assumptions broadly agree with the patterns seen in fishes: short, deep fishes tend to have short, deep heads, and elongate, shallow-bodied fishes typically have elongate, shallow heads (Figure 2, see also [46,47]: Figure 1). This suggests that head and body proportions may be strongly correlated in fishes, and hence dimensions of the head and thoracic armor can be used to predict the length of arthrodires. Here, I estimate body size (total length) and body mass (weight) in *Dunkleosteus* using a broad sample of extant fishes as well as arthrodires for which complete remains are known. In addition, I discuss the implications of these length estimates on our knowledge of arthrodire body shape, fish body shape evolution, and the expansion of maximum vertebrate body size during the middle Paleozoic.

## 2. Materials and Methods

### 2.1. Institutional Abbreviations

**AA.MEM.DS**, Université Cadi Ayyad, Marrakech, Morocco; **ANSP**, Academy of Natural Sciences, Philadelphia, PA, USA; **AMNH FF**, American Museum of Natural History fossil fish collection, New York City, NY, USA; **CMNH**; Cleveland Museum of Natural History, Cleveland, OH, USA; **FMNH**, the Field Museum, Chicago, IL, USA; **FSBC**, Florida Biodiversity Collection, Florida Fish and Wildlife Research Institute, St. Petersburg, FL, USA; **LDUCZ**, Grant Museum of Zoology, University College, London, U.K.; **MNHM**, Musée d'Histoire Naturelle de Miguasha, Quebec, Canada; **MZL**; Musée Cantonal de Zoologie, Lausanne, Switzerland; **NHMUK**, the Natural History Museum, London, U.K.; **NMS**, National Museum of Scotland, Edinburgh, UK; **OSUM**, Ohio State University Museum of Biological Diversity, Columbus, OH, USA; **ROM**, Royal Ontario Museum, Toronto, Ontario, Canada.

### 2.2. Model Assumptions

Any model intended to estimate the length of large arthrodires such as *Dunkleosteus* must fulfill four major criteria:

1. The dataset must include a wide variety of fishes, including fishes spanning the possible range of body sizes for *Dunkleosteus*. This is necessary to avoid errors from data extrapolation, which if not controlled for can lead to errors in body size estimation [30,35,36]. Related to this, it is important to include taxa that phylogenetically bracket the extinct taxa of interest, in order to increase confidence in the applicability of the model [48]. For arthrodires, this phylogenetic bracket would encompass extant gnathostomes (chondrichthyans, osteichthyans), lampreys (Petromyzontiformes), and other arthrodires for which complete remains are known (Figure 3). Lampreys are not the closest relative to gnathostomes among jawless fish groups (cephalaspidomorphs are closer), but were chosen here because lampreys can be measured from modern specimens and thus be measured more precisely.

2. The model must accurately estimate body size in fishes regardless of phylogeny. If a model only predicts body length in one group of fishes like sharks or bony fishes but cannot be applied more broadly, it is unlikely to be accurate in arthrodires. Similarly, a measurement may strongly correlate with total length in fishes but different groups of fishes may follow different regression lines. If this is the case, an additional variable would be needed to adjust for clade-specific differences in slope and intercept. However, such a model would be almost useless for estimating body size in arthrodires, as the additional coefficients for arthrodires would be calculated based on a narrow subset of taxa spanning a limited range of sizes.

3. The model must accurately estimate total length in the few arthrodire taxa known from complete remains. If a method works for extant gnathostomes (which are universally regarded as more closely related to each other than to arthrodires; [5,49]) but fails to predict length in Arthrodira, it cannot be reasonably applied to *Dunkleosteus*. One potential issue is that most arthrodires for which complete remains are known are either coccosteomorphs (e.g., *Coccosteus*, *Millerosteus*, *Watsonosteus*) or more basal arthrodire lineages (*Africanaspis*, *Holonema*). *Amazichthys trinajsticae* is the only exception in this regard [21]. However, given the distribution of taxa considered in this study (Figure 3), if a model accurately predicts body length in lampreys, coccosteomorphs, basal arthrodires, *Amazichthys*, and extant jawed fishes it can be assumed it will also accurately predict body length in *Dunkleosteus terrelli* and other "pachyosteomorph" arthrodires (Dunkleosteoidea and Aspinothoracidi).

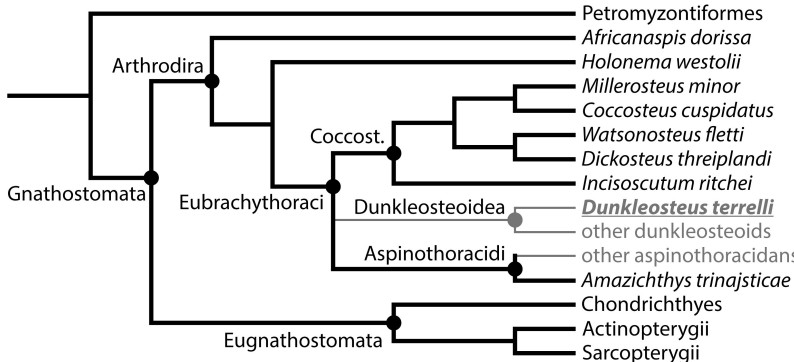

**Figure 3.** Phylogeny of taxa considered in this study. Taxa with known body lengths are listed in black, those where body length is unknown are listed in gray. Arthrodire phylogeny follows Zhu et al. [23], Boyle and Ryan [22], and Jobbins et al. [21]. Abbreviations: Coccost., Coccosteomorpha.

4. The anatomical proxy for total length must be measurable in fossils of *Dunkleosteus terrelli*. If a measurement is highly correlated with size but is not measurable in *Dunkleosteus* specimens (e.g., snout-vent length) or is based on anatomical landmarks that cannot be reliably recognized in arthrodires (e.g., prebranchial length, given the branchial region of arthrodires cannot be easily distinguished from the rest of the skull [50,51]), then it is useless for estimating the body size of *D. terrelli*.

*2.3. Data and Measurements*

Measurements of OOL, total length, and other body measurements of interest (Figure 4) were collected for *Dunkleosteus*, arthrodires known from complete remains, and a comparative sample of extant fishes. Measurements of *Dunkleosteus* were taken from specimens housed at the Cleveland Museum of Natural History. This museum contains four near-complete, three-dimensionally mounted specimens of *Dunkleosteus terrelli* (CMNH 7424, CMNH 6090, CMNH 7054, and CMNH 5768), spanning an ontogenetic series ranging from a juvenile (CMNH 7424) to a large adult (CMNH 5768). All four specimens pertain to single individuals with minimal restoration, and therefore there is little concern that their proportions could result from being composite individuals. Measurements of these specimens were collected either in person by the author or using 3D models from the University of Michigan UMORF (https://umorf.ummp.lsa.umich.edu/wp/specimen-data/?Model_ID=1336, accessed on 26 February 2022) or scans uploaded to MorphoSource (https://www.morphosource.org/, accessed on 15 August 2022) by the Cleveland Museum of Natural History. *Dunkleosteus* hypodigms housed at the American Museum of Natural History and The Natural History Museum, London, were also examined for comparison, though none of these specimens were complete enough to include in the analysis except for two individuals mounted in the AMNH's Hall of Vertebrate Origins, which could not be reliably measured (and may be composites; see [52]).

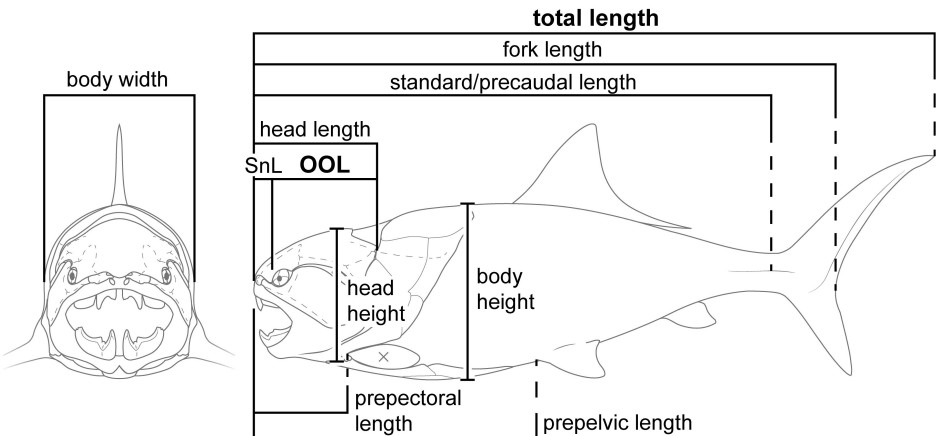

**Figure 4.** Reconstruction of a juvenile *Dunkleosteus terrelli*, showing the measurements used in this study. Abbreviations: SnL, snout length; OOL, orbit-opercular length.

Measurements of complete arthrodires were taken from specimens at the Field Museum of Natural History (FMNH), the Musée d'Histoire Naturelle du Parc National de Miguasha (MNHM), the National Museum of Scotland (NMS), and Royal Ontario Museum (ROM), as well as the previously published literature. This includes individuals of *Coccosteus cuspidatus* [15], *Dickosteus threplandi*, *Millerosteus minor* [53], *Watsonosteus fletti* (J. Newman, pers. comm.), *Plourdosteus canadensis* [19], *Africanaspis dorissa* [17], *Amazichthys trinajsticae* [21], *Holonema westolli* [16,54] and *Incisoscutum ritchei* [20,55]. Measurements for these taxa were collected either from published reconstructions or complete specimens, depending on which was available. For *I. ritchei*, most of this taxon's anatomy is known but the caudal fin is missing, as is the case for nearly all eubrachythoracid arthrodires from the Gogo Formation ([56]; Trinajstic pers. comm.). This prevents the otherwise spectacularly preserved Gogo arthrodires [55,57,58] from being used to estimate the length of *Dunkleosteus*. For *Incisoscutum*, because the anatomy of this arthrodire is otherwise completely known, caudal fin length was estimated for this species assuming a similar proportion between precaudal length and total length as *Coccosteus cuspidatus*, similar to Engelman [33]. For *Millerosteus minor*, the dimensions of this taxon were calculated using a composite of the specimen described in Desmond [53] (C.369 of that study, now catalogued as LDUCZ-V998) and an undescribed specimen under study by M. J. Newman.

LDUCZ-VPP8 is missing most of its tail whereas the Newman specimen is complete but has a poorly preserved head, hence the use of a composite specimen.

Only arthrodiran placoderms were considered in the present study. Although some other placoderm clades, such as ptyctodonts, rhenanidans, and antiarchs, are known from complete remains, these groups show highly specialized body plans that differ from the fusiform body plan seen in most arthrodires. For example, most ptyctodonts are macruriform and have "whip-like" caudal fins similar to chimaeroids [59,60], rhenanidans are dorsoventrally flattened similar to angel sharks [61], and antiarchs have a tadpole-shaped body plan with a box-like thorax and jointed, crab-like pectoral fins difficult to analogize to any animal alive today [61,62]. These unusual body plans might distort comparisons of body proportions between arthrodires and extant fishes, especially as they are correlated with natural history traits (e.g., a macruriform body plan or benthic habits) known to produce extreme body shapes in living fishes. It is possible OOL may accurately estimate length in other placoderm groups, but given the focus of the current study on arthrodires considering these taxa is beyond the scope of this study.

For several analyses of body dimensions in arthrodires (namely, percent snout length and length–weight patterns across Arthrodira), several additional taxa represented by complete armors but incomplete/unknown post-thoracic remains were added to the analysis. This includes several well-known reef-dwelling taxa from the latest middle Devonian Gogo Formation of Australia (e.g., [63–65]) and several pelagic contemporaries of *Dunkleosteus* from the latest Famennian Cleveland Shale. For life habitus, the Cleveland Shale taxa were all assumed to be pelagic. The Gogo Formation eubrachythoracid arthrodires were treated as demersal due to being considered reef-dwelling taxa [56] with the exception of the camuropiscids, which are considered active, nektonic animals [56,66] and thus treated as neritic. Future analysis may find some of the Gogo arthrodires treated as demersal here (e.g., *Eastmanosteus*, *Incisoscutum*) were actually neritic, but it is clear these taxa are not benthic or pelagic and it was necessary to code them as belonging to one group for the purpose of the statistical analysis. Treating the majority of the Gogo arthrodires as demersal is the more conservative of the two options for now. Similarly, all arthrodires except *Amazichthys* (see Discussion) were treated as having a fusiform body shape, though the author suspects *Gymnotrachelus* and the camuropiscids (e.g., *Rolfosteus*) might eventually be identified as exhibiting elongate or "semi-elongate" body plans akin to some mackerels or triakid sharks, based on the proportions of their head and body armor.

Data for extant fishes were collected from the previously published literature, as well as extant fish specimens in the collections of the Cleveland Museum of Natural History (CMNH), Ohio State University Museum of Biodiversity (OSU), and the Florida Biodiversity Collection at the Florida Fish and Wildlife Research Institute (FSBC). Special attention is drawn here to the photos uploaded to FishBase [67] by Randall [45], for which total, fork, and/or standard length (if available) were reported by J. E. Randall, but all other measurements were collected by the author by using the measurement program in Adobe Acrobat. Data for extant fishes were collected as available, though an effort was made to collect data from a phylogenetically diverse array of fish taxa (especially early-diverging lineages of major clades) and include most major fish species expected to be comparable in size to *Dunkleosteus terrelli* (e.g., large sharks, thunnins, megalopids, billfishes, etc.).

In total, this comparative sample includes data from 3159 distinct occurrences and 969 taxa, including Actinopterygii (2210 observations, 770 species), Chondrichthyes (568 observations, 181 species), Petromyzontiformes (358 observations, 8 species), Sarcopterygii (16 occurrences, 3 species), and Arthrodira (17 observations, 10 species), not including arthrodires for which total length was unknown. Individual specimens were used whenever possible, but in some cases observations represent sample averages due to the way they were reported in the literature. Because measurements in the ichthyological literature are typically reported as proportions relative to standard, total, or fork length, including sample means from multiple individuals is not expected to bias the results by

using numeric measurements of individuals of different sizes. Measurement definitions largely follow Hubbs et al. [68] and Compagno [69], with some modifications.

OOL is defined here as the length from the anterior margin of the bony orbit to the posterior margin of the gill region along the anteroposterior axis of the body. This is equivalent to the common ichthyological definition for head length (length from tip of the rostrum to posterior end of the gills; [68,69]) minus the preorbital/snout length. Definitions for the posterior boundary of the gill chamber differ slightly between groups due to differences in external gill anatomy. In bony fishes and holocephalians this landmark is the posterior margin of the operculum (Figure 1A), whereas in elasmobranchs it is the posterior margin of the terminal (usually fifth) gill arch (Figure 1B). In arthrodires OOL was measured to the level of the cranio–thoracic joint (Figure 1C,D), as in this group the gill cover and cheek bones are incorporated into a single unit (the suborbital plate) located more or less ventral to the neurocranium (rather than posteroventral as in bony fishes or chondrichthyans). Thus, the ventral margin of the brain case and the posterior margin of the gill chamber (as marked by the postbranchial lamina; [50,64,70]) are roughly at the same level as the cranio–thoracic joint in most "pachyosteomorph" arthrodires (Figure 1D). In coccosteomorphs, the neurocranium may even extend slightly posterior to the gills (Figure 1C), thus using the cranio–thoracic joint as the posterior landmark is necessary to encompass the entire head. Despite these varying definitions (mostly driven by the presence/absence of an external gill cover), the posterior landmark for OOL represents a biologically homologous point among fishes. For arthrodires, OOL was measured with the head in a natural resting position, rather than measured along its greatest midline length (oblique to the anteroposterior axis), as in Miles and Dennis [64] and subsequent studies. Measuring head length/OOL in this manner would make it impossible to compare measurements from arthrodires with other fishes and would cause difficulties in arthrodires with substantial nuchal embayments, like *Dunkleosteus*.

Several groups of fishes (i.e., serranids, centropomids) have an opercular flap, a fleshy extension of the opercular margin that expands the overall dimensions of the head in lateral view. However, direct examination of preserved fishes with opercular flaps finds the dimensions of this flap do not correlate with the posterior end of the gill chamber. Instead, the gill chamber ends significantly anterior to the opercular flap, approximately at the level of the bony operculum, and the branchiostegal membrane typically does not attach to the opercular flap but attaches to the end of the bony operculum. As a result, including the opercular flap as part of OOL violates the assumption that this measurement spans the neurocranium and gill chamber and results in substantial overestimates of total length, even in taxa like serranids that already show apomorphic shifts in this relationship. As a result, when possible, OOL (and head length) was measured omitting the length of the fleshy opercular flap (e.g., to the posterior base of the bony operculum). Notably, this could not be done in all of the data from the prior literature, as head length is typically measured including the opercular flap [71], and not all of the studies included figures that allowed head length and OOL to be remeasured subtracting the opercular flap.

Body length in this study was measured as total length: the length from the anterior tip of the snout to the posteriormost extent of the caudal fin (in natural position, when possible). This is the way length is typically measured in studies of extant chondrichthyans, sarcopterygians, and lampreys, as well as in most paleontological studies. However, this is notably not how body length is typically defined in actinopterygians. In these taxa body length is measured as "standard length": the length to the end of the spinal column excluding the caudal fin and part of the hypural plate [68]. This meant that measurements as reported in many studies of actinopterygians could not be used, as in most studies standard length is reported but caudal fin measurements are not. However, total length is used for actinopterygians here to make them comparable with arthrodires and other fishes.

In cases where it was possible to measure total length directly, total length was measured with the caudal fin in a natural position rather than it being depressed or stretched out as in some studies (see, e.g., discussion in [72]). However, this could not be accounted

for in all of the literature data, as these studies often do not note what method they used to measure length. The existence of significant inter-observer variation in how body length is measured in fishes (i.e., standard versus total length, definition of caudal peduncle, caudal fin in natural position or stretched out) is a known issue in ichthyology [72,73], but it is something that cannot be avoided here due to the nature of the data.

To test whether the relationship between OOL and total length was potentially biased by clade-specific patterns in body shape, taxonomic information was included as an additional categorical variable in some of the analyses. The classification of fishes in this study follows the general consensus phylogeny of non-tetrapod vertebrates (Figure 3, see also [74]) with the exception of teleosts (especially for Acanthomorpha), in which relationships are not entirely settled. Teleost classification largely follows Betancur et al. [75] and Hughes et al. [76]. Although there is still controversy over whether "Placodermi" is para- or monophyletic (see [1–5]), because only arthrodires are considered here differences in the classification of "placoderms" are not expected to affect the study. Despite controversies over placoderm monophyly, virtually all authors agree that Arthrodira is positioned basal to Chondrichthyes + Osteichthyes (again, see [1–5]).

Fish body shape and proportions are also influenced by a number of ecological factors, including life habits. To account for the potential influence of life habits on length estimates in *Dunkleosteus*, a character for life habits was included based on the general life mode of the taxon in the previously published literature. Life habits were simplified into four broad-scale categories for easy analysis: benthic, demersal, nektonic (i.e., nektonic but associated with coastal environments or the substrate), and pelagic (i.e., nektonic but living in open waters). *Dunkleosteus* was treated as pelagic following Carr [11]. The other arthrodires for which complete remains are known are all generally regarded as demersal taxa, except for the pelagic *Amazichthys* [21] and the benthic *Holonema* [16,54] and *Millerosteus* [53]. Similarly, a categorical variable with the levels fusiform, compressiform, elongate, anguilliform, or macruriform (see Figure 2, also Supplementary File S3) was included to investigate variation in OOL among fishes with different body shapes.

### 2.4. Statistical Analysis

All analyses and statistical calculations were performed in R 4.2.1 [77]. No additional packages were used to perform the statistical analyses in this study, though others such as the *tidyverse* suite [78] and the packages *broom* [79], *cowplot* [80], *e1071* [81], *ggstar* [82], *ggtext* [83], *gridExtra* [84], *kableExtra* [85], *magick* [86], *magrittr* [87], *openxlsx* [88], *readxl* [89], *rlang* [90], and *scales* [91] were used in data visualization. A knitted .html file of all analyses can be found in the Supplementary File S3. The original R code in .rmd format can be downloaded using the tab labelled "Code" at the very top of the document and rerun to replicate all statistical analyses performed in this study.

Models were calculated using both individual data points (i.e., multiple points per taxon) and species averages. This was done for several reasons. Species averages are the standard for interspecific size-estimation models in many organisms, such as mammals [92]. This is partly because mammals have determinate growth and easily identifiable markers of somatic maturity (i.e., tooth eruption, fusion of growth plates). By contrast, fishes show indeterminate growth and most ichthyological studies make little distinction between juvenile and mature forms (e.g., most morphometric studies of sharks are based on sexually immature individuals, see Garrick [37] for an example). Although an attempt was made to filter out obvious juveniles before performing this analysis, many studies do not provide ontogenetic information on their specimens (and in many fishes sexual maturity cannot be verified without dissection) and often mix immature and adult individuals (e.g., many studies of pelagic actinopterygians include juvenile specimens but do not mention what proportion of their sample represents mature adults or provide measurements for only adults specimens; [93–97]). This means maturity had to be inferred, if possible, through context. This means in a species-average model, unrecognized juveniles have the potential to downweight the mean total length of the taxon. This is a problem for estimating body size

in megafaunal taxa like *Dunkleosteus*, as data from as many large-bodied taxa as possibly are necessary to minimize extrapolation error [30]. Including juvenile specimens results in a smaller average size for a species and thus effectively reduces the number of large bodied taxa in the sample.

A related issue is that there is an uneven distribution of taxa and specimens at larger body sizes. The largest (>2 m) extant fishes mostly pertain to one of three clades: sharks (Elasmobranchii), billfishes (Istiophoriformes), and tunas (Thunnini). Large sample sizes are available for billfishes (Istiophoriformes), specifically the genera *Kaijikia* [93] and *Tetrapturus* [94,95]. However, the large sample sizes for these taxa and their specialized, elongate body shape drowns out the morphological signal from other groups and results in the best-fit line being significantly dragged upward at larger body sizes (i.e., increasing length estimates in *Dunkleosteus* by 20 cm or so). However, while species-averages avoid the issue of over-representation by Istiophoriformes, they result in an over-representation by certain groups of sharks, specifically *Carcharhinus* spp., which comprises 36 species [98] that are all similar in body shape but are considered distinct at the species level [37]. Thus, either method results in over-representation of some taxonomic group, and results of both individual data points and species averages are reported.

All length estimates were performed via ordinary least squares (OLS) regression, rather than phylogenetic generalized least squares. There are two main reasons for this. First, fish phylogeny, specifically the relationships within Acanthomorpha, is still not completely resolved [75,76]. Although species-level phylogenies are available for sharks (e.g., https://vertlife.org/phylosubsets/), reliable species-level phylogenies are not available for many of the bony fish taxa considered here. Many of the taxa for which extensive morphometric data are available are either rare or recently described taxa, and thus often lack good phylogenetic information. For example, using the *fishbase* package [99], roughly one-third of the actinopterygian taxa considered here do not have phylogenetic information in Rabosky et al. [100].

Second, most available PGLS packages, particularly those in R, currently do not provide methods for incorporating phylogenetic information when predicting values for new taxa [101–103], though methods for doing so have been proposed in the literature [104]. PGLS packages as they currently exist estimate values without considering the effect of phylogenetic non-independence, which is equivalent to treating new taxa as located at the root of the entire tree (see [102] and discussion therein). This often results in PGLS returning lower prediction accuracy [36,101] and wider prediction intervals [105] than OLS, even though incorporating phylogenetic information should produce more precise estimates of body size [104]. Given the very strong correlation between OOL and total length recovered here, the use of PGLS is unlikely to significantly alter the results of this study, though incorporating phylogenetic signal would likely improve model precision. Predicting the length of *Dunkleosteus* using phylogenetic comparative methods is a future goal and an obvious next step from this analysis, but given the length of the present manuscript it is considered beyond the scope of the current study.

Model accuracy was measured using $r^2$ values, percent error (%PE), percent standard error of the estimate (%SEE), Akaike information criterion (AIC; [106]), Bayesian Information criterion (BIC; [107]), and log likelihood (logLik). The $r^2$ value, also known as the correlation coefficient, is the traditional support statistic of choice in the ichthyological literature, but has several key statistical deficiencies that are covered by the other methods used here. Specifically, $r^2$ cannot directly determine how accurate the resulting model is at predicting new values [108,109]. The $r^2$ value is also very sensitive to the range of sizes spanned by the data. As the magnitude of sizes spanned by the data increases $r^2$ unilaterally increases, even when prediction accuracy is very low [40,109]. Thus, even models where prediction error is so high as to make the model uninformative can have an $r^2$ greater than 0.99 if data spans a wide enough range of sizes [109]. This problem is magnified in log-transformed datasets (i.e., most analyses of biological data), because the spread of the data is compressed by log-transformation and $r^2$ tends to overestimate

confidence in prediction accuracy. The $r^2$ value measures the correlation between the log-transformed data, when the value of interest is how accurately the model predicts data on the original untransformed scale. Thus, %PE and %SEE, which measure how well the model predicts new data on a log-detransformed scale, were used as the primary measures of model accuracy.

When estimating unknown values using a log–log model, error (known as log-transformation bias) is introduced because data are predicted on a unitless, log-transformed scale but must be detransformed back into units of interest [110,111]. Several correction factors have been proposed to correct for log-transformation bias, including the quasi-maximum likelihood estimator [110,111], smearing estimate [110–112], and ratio estimator [110,111,113]. To correct for log-transformation bias, these three correction factors were calculated for the present data, their average was calculated, and the predicted values were multiplied by this averaged correction factor to produce corrected length estimates, following the methodology of previous studies [101,114].

The accuracy of these regression models when applied to arthrodires was further tested by applying them to several arthrodire taxa for which complete remains are known (see "Data and Measurements", above). The idea beingthe proportions of these smaller arthrodires would be used as an independent test for the present analysis: if the equations accurately estimated length in smaller, complete arthrodires, it is more likely they will accurately estimate length in *Dunkleosteus* as well. The use of smaller arthrodires as proxies for *Dunkleosteus* has been criticized [12], but in this case if a prediction method does not work on arthrodires at all the chances of it being applicable to *Dunkleosteus* are low.

Following evaluation of the model, total length was estimated for specimens of *Dunkleosteus terrelli* at the Cleveland Museum of Natural History in which OOL could be measured. Special focus was given to CMNH 5768, which is a large, likely adult individual of *Dunkleosteus terrelli* (Figure 5) and the individual that serves as the basis for most casts of *D. terrelli* displayed throughout the world. Thus, this individual can be thought of as a representative adult specimen of *D. terrelli* (with the other three individuals representing juveniles or young adults based on unpublished observations; Engelman, pers. obs.).

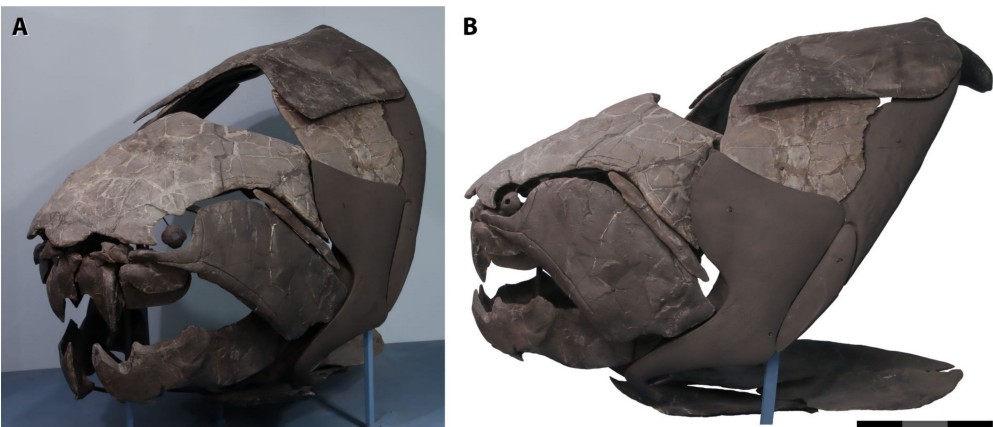

**Figure 5.** CMNH 5768, the largest complete individual of *Dunkleosteus terrelli* in oblique left lateral (**A**) and left lateral (**B**) view. Some of the thoracic plates are reconstructed but the skull and ventral armor are entirely real. The proportions of the reconstructed plates closely resemble other specimens of *Dunkleosteus*. Note (**B**) is taken at a slightly oblique angle because a camera could not be placed in perfectly lateral view. Scale = 30 cm, but only applies to (**B**), (**A**) is scaleless.

A reconstruction of *Dunkleosteus*, largely modeled after CMNH 5768, was made to help convey the results of this study. The anatomical details of this reconstruction (e.g., lips or no lips, visible armor, body shape) are the subject of a manuscript currently in preparation [115]. However, for the purposes of the present study it is important to note the bony anatomy of this reconstruction was drawn directly from CMNH 5768, either from

a digital model of this specimen on University of Michigan UMORF (https://umorf.ummp.
lsa.umich.edu/wp/specimen-data/?Model_ID=1336, accessed on 26 February 2022) or
from direct observations of CMNH 5768 and other *Dunkleosteus* material at the Cleveland
Museum of Natural History. Thus, the dimensions of the armor in the reconstruction reflect
the actual dimensions of CMNH 5768, rather than being approximations.

### 2.4.1. Estimating the Length of the Largest Dunkleosteus

The largest known individual of *Dunkleosteus terrelli* (CMNH 5936) is an isolated
partial inferognathal, and thus OOL cannot be measured in this specimen. However, a
regression equation can be created between OOL and measurements of the inferognathal
from more complete individuals of *D. terrelli* to estimate OOL in CMNH 5936. This
predicted OOL can then be used to estimate the total length of this individual. This is the
same method Ferrón et al. [12] used to estimate the total length of CMNH 5936, except
using OOL as the intermediate size proxy rather than upper mouth perimeter. Regression
equations were created between OOL and the five measurements of the inferognathal
used by Ferrón et al. [12] to estimate the length of CMNH 5936 (Figure 6). However, in
practice only two of the five measurements proposed by Ferrón et al. [12] reliably correlated
with size: length of the oral region of the inferognathal (JM5 of [12]) and height of the
inferognathal at the posterior cusp (JM3 of [12]). The other measurements showed too
much intraspecific variation to reliably estimate body size (see Supplementary File S3:
Section S14.1). The morphology of the biting surface is highly variable in *Dunkleosteus*,
with the height and position of the median accessory cusps being particularly plastic (see
also [116,117]). This, in turn, affects the reliability of the correlation between the remaining
measurements (JM1, JM2, and JM4) and size.

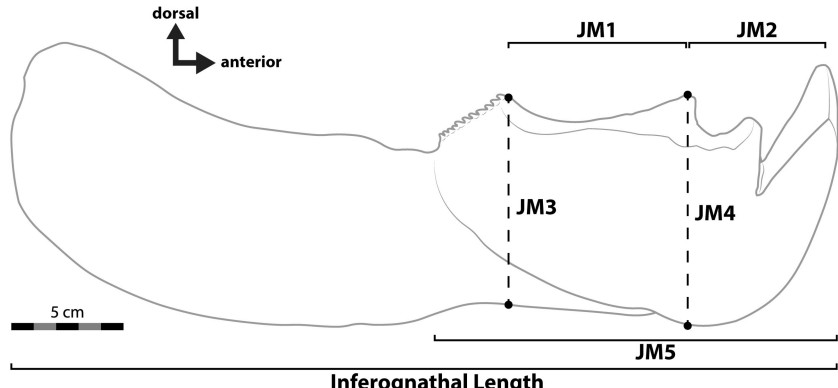

**Figure 6.** Inferognathal of *Dunkleosteus terrelli* (CMNH 5698, reversed), showing the measurements of
this bone used to estimate the length of CMNH 5936. Definitions of JM1-JM5 follow Ferrón et al. [12].

Associated OOL and inferognathal dimensions were measured for 11 specimens of
*Dunkleosteus terrelli* in the collections of the CMNH (see Supplementary File S3: Table
S14). These specimens all represent single individuals, rather than composite specimens.
Measurements from left and right inferognathals were included separately when possible to
control for potential taphonomic distortion. OOL in CMNH 5936 was estimated using the
height of the inferognathal at the posterior cusp and the length of the oral region. This was
done using non-log-transformed equations to avoid log-transformation bias and the wide
prediction intervals of log-transformed models. The specimens considered here spanned
a narrow enough range of sizes that log-transformation was unnecessary. This predicted
OOL was then used to estimate the total length of CMNH 5936.

### 2.4.2. Body Mass of Dunkleosteus

The body mass (dry weight in air) of *Dunkleosteus terrelli* was estimated using two
methods. First, body mass was estimated using a multivariate ellipsoid model. This is an

extrapolation of the method used by Ault and Luo [118] to estimate body mass in Atlantic tarpon (*Megalops atlanticus*), with modifications to make it more suitable for estimating body mass across all fishes. The body plan of fishes can best be modeled as an ellipsoid, and therefore if body length (either as total length, fork length, or precaudal length, depending on the parameters of the study) and body girth are known it is theoretically possible to approximate body mass. For this study, precaudal length was used as the measurement of body length, as it ignores variation in caudal fin shape among taxa. However, an additional parameter was also added to account for taxa with a heterocercal caudal fin, in which the axial skeleton extends into the caudal fin and hence would be expected to contribute more to total mass. In *Dunkleosteus*, body girth can be measured directly from the thoracic armor of mounted specimens, whereas precaudal length can be calculated from estimated total length as the two values are strongly correlated across fishes ($r^2$ = 0.996). Precaudal length and girth were used to create a regression model estimating the body mass of *Dunkleosteus*. This model takes the form of:

$$\text{body mass} = (\text{girth}^2) \times \text{length} + \text{girth} \times \text{length} + \text{girth}^2 + \text{girth} + \text{length} \\ + \text{length} \times \text{head length} + \text{``presence of swim bladder?''} + (\text{``is tail heterocercal''} \qquad (1) \\ \times \text{caudal fin length})$$

in which "presence of swim bladder" and "is tail heterocercal" are binary categorical variables that have a significant influence on the results. Additional details of the assumptions of this model, including a brief overview of the methodology of Ault and Luo [118], can be found in Supplementary File S3.

Although many of the fish specimens used in this study had recorded weight data, many did not. For these specimens, body mass was estimated using published length–weight equations for these species (or closely related congeners of similar shape if no published equation was available). For a few taxa, the Bayesian model of Froese et al. [119] had to be used because weight data were unavailable for these species. However, the majority of fishes in this study (>95%) either had associated weights or their weight could be estimated through species-specific length–weight equations. Because of previous concerns regarding statistical errors with reported length–weight models in fishes [120], the models used were vetted to use the best-fitting models possible (using criteria of Froese [121]). When length–weight equations were only available for separate sexes and the gender of the fish in question was unknown, equations for males were used, as higher length–weight ratios in female fishes are often related to the production of eggs or developing young and thus not expected to be due to differences in body proportions. No attempt was made to record body mass by directly weighing fluid-preserved specimens due to concerns of how preservation might bias these estimates (i.e., loss of body fluids or viscera, or an increase in weight due to preserved fishes absorbing alcohol [122]). Further information as to which specimens had their body mass estimated and how can be found in Supplementary File S1.

Second, body weight was estimated using the volumetric model created by Mollet and Cailliet [123] for *Carcharodon carcharias*. This equation takes the form of:

$$\text{body mass} = 45.98 \times (\text{precaudal length} \times \text{girth}^2)^{0.9267} \qquad (2)$$

Although *Carcharodon* and *Dunkleosteus* likely do not have identical body shapes (see below), *Carcharodon* represents one of the best extant models to estimate the weight of *Dunkleosteus* because of its similar size (reducing extrapolation errors; [30]), similar inferred life habits (both are pelagic predators; [11]) and a potentially similar body shape (see [12,115], and below). Additionally, because the formula of Mollet and Cailliet [123] is calculated using length and girth, it is slightly more robust to interspecific differences in body proportions between sharks and arthrodires (see below) than a simple length–weight equation.

Notably, these methods do not attempt to account for the presence of dermal armor when estimating the body mass of *Dunkleosteus*. Thus, these estimates may be conservative and potentially slight underestimate the true value. To test the effect the bony armor of

*Dunkleosteus* might have on weight estimates, the volume of the armor was calculated from a 3D surface scan of one of the mounted specimens of *Dunkleosteus* considered in this study (CMNH 6090). The mass of the armor was calculated by multiplying its volume by the density of whole bone (1.2–1.3 g/cm$^2$; [124,125]). Using these values for bone density, which were calculated based on terrestrial tetrapods, seems reasonable, given histological studies of arthrodire plates show cancellous and cortical layers similar to terrestrial tetrapods [126]. This is distinctly unlike either the pachyostotic bones of most aquatic tetrapods [127,128] or the acellular bone typical of euteleosts [129,130]. This method was used to provide a conservative estimate of by how much the body armor of *Dunkleosteus* would be expected to bias body mass estimates based on non-armored taxa.

## 3. Results

### 3.1. Results of Model

#### 3.1.1. OOL in Extant Fishes

Orbit-opercular length scales nearly isometrically with total length, with a log-transformed slope of 0.947 (Figure 7). Although the relationship is near isometric, the presence of non-normality and heteroskedasticity in the residuals of the untransformed model with a few very large values suggests log-transformation is necessary (Supplementary File S3: Section S5.4). The diagnostic plots of the natural log-transformed model show normally distributed residuals, little heteroskedasticity, and no highly influential outliers (Supplementary File S3: Section S5.3). Prediction error (PE$_{cf}$) for the model is relatively low (17.55%). That is, on average OOL without any additional parameters will predict the body length of a given fish within +/−17.55% of the actual value (Table 1). 88% of all sampled fishes have estimated lengths within +/−33% of their actual value, whereas roughly 2/3 have their length estimated within +/−20% (Supplementary File S3: Section S5.7). Taxa outside this interval are mostly those with very extreme body shapes that are not typical for fishes (e.g., highly anguilliform taxa).

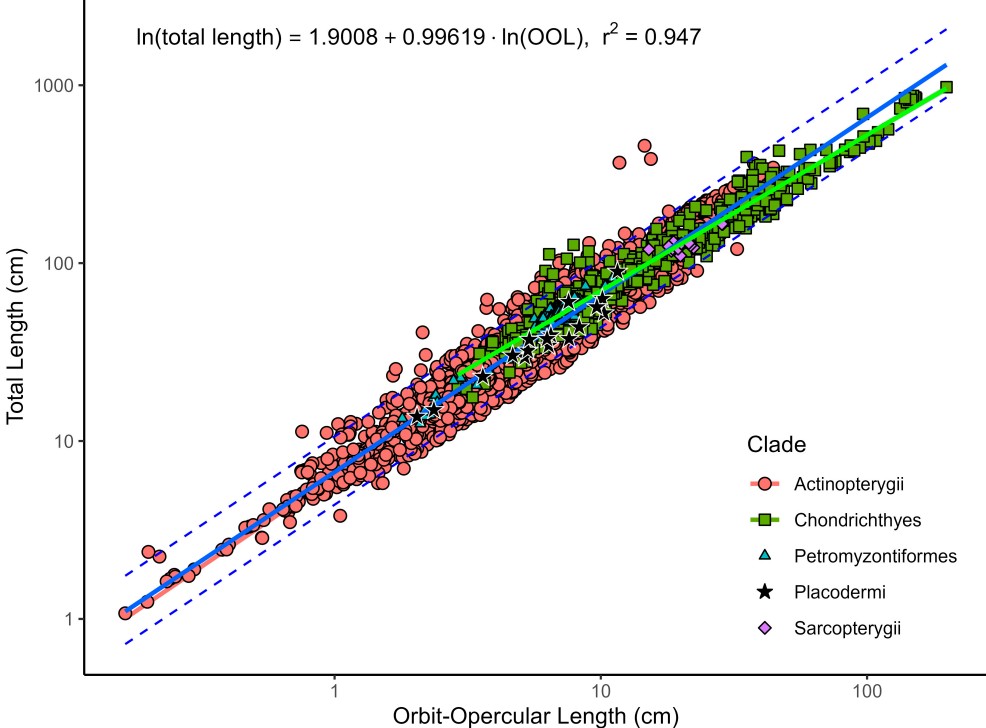

**Figure 7.** Plot of log$_{10}$ orbit-opercular length against log$_{10}$ total length in fishes. Solid and dashed blue lines represents best-fit regression line and 95% confidence intervals for all species, respectively.

Overall, aspect ratios of the head and body are highly correlated across fishes ($r^2 \sim 0.8$; Supplementary File S3: Section S3). In other words, the general observation that short fishes have short, deep heads and elongate fishes have elongate heads is generally true (see also [46]). However, the model shows residual variation significantly correlated with body shape ($t = 287.22$, $p < 0.001$; Supplementary File S3: Section S7.2.3). Anguilliform and macruriform fishes show the greatest underestimates of body length, elongate-bodied fishes slightly less so, fusiform fishes show residuals close to zero, and compressiform fishes show positive residuals and overestimates of body length (Supplementary File S3: Figure S7). However, these differences are only differences in intercept, not slope (Supplementary File S3: Section S7.2). Slopes of different shape categories are non-significantly different except for elongate fishes, which may be due to under-sampling of small-bodied elongate fishes.

**Table 1.** Regression equations and support statistics for some of the best-fitting models. All equations reported here are for models using individual specimens, not species averages. Additional statistical information, including models fitted using species averages, can be found in Supplementary File S3. Note more complex multivariate equations are listed in Supplementary File S3 due to space constraints. Abbreviations: TL, total length; OOL, orbit-opercular length; HDL, head length; SNL snout length.

| Model | N | Equation | $r^2_{adj}$ | AIC | BIC | %PE | CF | %PE$_{cf}$ | %SEE |
|---|---|---|---|---|---|---|---|---|---|
| All species | 3169 | Ln(TL) = 0.9962 × Ln(OOL) + 1.9008 | 0.947 | −463 | −445 | 17.83 | 1.019 | 17.55 | 25.21 |
| Fusiform and elongate taxa | 2660 | Ln(TL) = 0.9836 × Ln(OOL) + 1.9622 | 0.962 | −1164 | −1147 | 15.38 | 1.011 | 15.26 | 21.44 |
| With shape as covariate | 3398 | See Supplementary Information | 0.974 | −2846 | −2785 | 12.10 | 1.009 | 12.03 | 17.23 |
| Fusiform species only | 1741 | Ln(TL) = 0.9713 × Ln(OOL) + 1.9121 | 0.980 | −1562 | −1545 | 11.98 | 1.008 | 11.88 | 16.69 |
| Including body depth as covariate | 2845 | See Supplementary Information | 0.950 | −761 | −737 | 16.26 | 1.023 | 16.17 | 23.56 |
| Including snout length as covariate | 3169 | Ln(TL) = 0.7482 × Ln(OOL) − 0.2301 × Ln(SNL) + 2.124 | 0.961 | −1451 | −1426 | 14.82 | 1.018 | 14.62 | 21.21 |
| Pelagic species only | 638 | Ln(TL) = 0.9677 × Ln(OOL) + 2.0373 | 0.953 | −256 | −242 | 16.65 | 1.009 | 16.56 | 21.83 |
| Fusiform and elongate non-acanthopterygians | 2394 | Ln(TL) = 0.9902 × Ln(OOL) + 1.8915 | 0.960 | −687 | −670 | 16.47 | 1.017 | 16.26 | 23.3 |
| Sharks only | 540 | Ln(TL) = 0.8852 × Ln(OOL) + 2.1809 | 0.962 | −544 | −531 | 11.57 | 1.012 | 11.51 | 15.69 |
| With shape, allowing variable slope for Chondrichthyes | 3169 | See Supplementary Information | 0.971 | −2310 | −2237 | 12.45 | 1.015 | 12.40 | 18.27 |
| Head length | 3169 | Ln(TL) = 0.9717 × Ln(HDL) + 1.5688 | 0.963 | −1579 | −1561 | 14.55 | 1.018 | 14.37 | 20.75 |

In other words, while head and body proportions in fishes covary far more closely than would be expected by chance, there is still unaccounted residual variation that is correlated with overall body shape. This suggests a non-linear pattern of variation in the body shape of fishes with the trunk elongating or compressing at an accelerated rate relative to the head, despite their proportions otherwise being closely associated. This results in a simple linear relationship being unable to completely describe variation in the relationship between OOL and body length. However, this biasing effect is most pronounced for species with extreme body plans (i.e., anguilliform and highly compressiform/discoid fishes), whereas the biasing effect is minimal for fusiform taxa. Therefore, it should be relatively safe to apply this model to arthrodires, which virtually all show fusiform body shapes.

The relationship between OOL and total length in fishes can effectively be described with a single model, with lampreys (Petromyzontiformes), arthrodires, chondrichthyans, sarcopterygians, and actinopterygians all falling along a single regression line (Figure 7). Because the taxon of interest (*Dunkleosteus terrelli*) is phylogenetically bracketed by the taxa included in this model, namely the jawless lampreys, the Eugnathostomata, and other arthrodires (Figure 3), this means that the present model should accurately predict length in *Dunkleosteus*.

When testing for differences in slope between clades, at first glance sharks appear to have a slightly different slope than bony fishes ($t = 11.621$, $p < 0.001$; Supplementary File S3: Section S5.9). However, further investigation finds this is a consequence of the non-random distribution of body shapes among sharks with respect to body size. Small shark taxa (e.g., Dalatiidae, Scyliorhinidae, Hemiscyllidae, Parascyllidae) are predominantly benthic species

and almost invariably have very elongate body shapes, whereas the largest sharks are pelagic taxa (e.g., Lamnidae, Megachasmidae) with slightly shorter, more tapered bodies relative to their heads, even compared to other fusiform fishes. This can be seen in the fact that other large sharks which do not show a lamnid-like body shape, such as hexanchiids, show residuals much closer to the best-fit line for all fishes, whereas etmopterids, which are elongate-bodied but pelagic, tend to show OOL proportions closer to lamnids. The patterns seen in sharks resemble patterns of variation in bony fishes, where elongate-bodied fishes tend to show systematically higher residuals (i.e., underestimates of length), whereas pelagic fishes with tapered trunks (i.e., many scombrids) often show overestimates of length similar to lamnids. If using species averages (i.e., minimizing the effect of juveniles and multiple observations per species) and adding categorical body shape as a covariate for intercept, slopes no longer differ between Chondrichthyes and Actinopterygii (t = −0.943, *p* = 0.320). This further suggests the differences in slope between sharks and other fishes are driven by the non-random patterns of body shape variation within sharks, rather than, say, differential growth of the head between different fish clades.

It is possible that there could even be a non-linear allometric relationship [131] between OOL and total length, as has been demonstrated for skull length and body mass in mammals [36,101] and possibly crocodilians ([132]; A.L. Paiva, pers comm.). This could also potentially explain the seeming difference in slopes between chondrichthyans and bony fishes, as sharks and bony fishes do not overlap in large regions of their respective size distributions [133]. However, this also makes it difficult to test whether these patterns are driven by a single, universal non-linear allometric relationship within fishes or different scaling relationships for different fish groups, due to phylogeny and size being confounded.

There seems to be a slight negative ontogenetic allometry between OOL and total length, with fish species (specifically actinopterygians) tending to show more negative residuals with increasing body size. That is, juvenile fishes have proportionally larger OOL relative to their body and "grow into" their adult proportions, with the largest adult fishes often fitting theoretical expectations of OOL-body proportions better than juveniles. In actinopterygians for which ontogenetic series could be measured firsthand (e.g., *Micropterus dolomieu*, *Hiodon tergisus*) there was a clear negative intraspecific allometry for OOL (log slope = ~0.9, Supplementary File S3: Figure S5). This also seems to be present in larger-bodied actinopterygians like *Acanthocybium*, *Thunnus*, and *Kajikia* (Supplementary File S3: Figure S5). A similar pattern appears to be present in the lamprey *Geotria* (data from Baker et al. [134]), though the allometric slope is not as extreme. However, this pattern is not present in elasmobranchs for which large sample sizes are available. The distribution of this pattern suggests that it may be related to the presence of a larval stage and hence more extreme shifts in body shape throughout ontogeny. OOL is also isometric with body size (log slope = 0.978; Supplementary File S3: Section S5.10.4) for the sarcopterygian *Eusthenopteron foordi* (data from Schultze [135]), which is also considered to lack a larval stage [136]. Although this pattern is reported here in the interests of transparency it is not expected to greatly bias the estimated lengths of arthrodires. This is because the model primarily focuses on adult fishes and if the pattern is related to the presence of a larval stage it is unlikely to be present in Arthrodira, which lack a larval stage [58].

Further examination of the residuals suggests that hypertrophy of certain cranial structures may explain some of the remaining variation in the relationship between OOL and total length. These include the presence of large mandibular adductor chambers, as seen in *Pygocentrus*, *Serrasalmus*, *Piaractus*, and *Colossoma* compared to other serrasalmids, which results in OOL overestimating total length. The same is true of taxa with large branchial cavities, which include some active, pelagic fishes like thunnins and lamnids as well as suspensorial filter feeders like *Polyodon* or the sharks *Rhincodon*, *Megachasma*, and *Cetorhinus*. Similarly, some Cypriniformes seem to have smaller heads than expected, which may be driven by the loss of oral teeth and de-emphasis of the true jaws in food processing in these fishes [137]. The present dataset cannot easily parse this variation, but given *Dunkleosteus terrelli* is thought to be an active pelagic animal with well-developed

jaw musculature [8,9,138] this suggests *D. terrelli* should have a larger head relative to body size, contrary to most prior reconstructions of the species.

The present data show some evidence that acanthopterygian fishes may have experienced an evolutionary shift towards proportionally larger heads (or rather, may have evolved truncated thoracic regions in association with the anterior shift of the pelvic girdle; [139]). Acanthopterygian fishes generally show slightly positive residuals (i.e., larger OOL relative to total length) relative to other fishes, with the exception of Carangiformes, Scombriformes, Istiophoriformes, and the highly elongate Beloniformes and Syngnathiformes. Some families, such as Serranidae and Holocentridae, show particularly high residuals. When comparing the species-average residuals of the OOL model against higher clade, using Chondrichthyes as the base level to contrast against due to its greater sampling and typically fusiform body shape, acanthomorph fishes have significantly larger OOL relative to body length (t = −4.669, $p < 0.001$; Supplementary File S3: Section S6.2). When including categorical body shape as an additional explanatory variable, acanthopterygians no longer differed significantly from other groups (t = −1.778, $p = 0.076$). However, when considering total head length, acanthomorphs had larger heads than other fishes (t = −8.757, $p < 0.001$) even when shape was controlled for (t = −7.767, $p < 0.001$). Given this result, an additional model was fit to test if excluding acanthopterygians, with their high taxonomic diversity and potentially shorter trunks, had a significant effect on length estimates of *Dunkleosteus*.

### 3.1.2. Outliers in the OOL Model

Given the extreme diversity of body shapes seen in modern fishes, it is unsurprising that some groups would fail to conform to the close relationship between OOL and body length found in the present study. However, examining these outliers in more detail provides insight into why the present model works so well in fishes, and why it would be expected to accurately predict length in *Dunkleosteus*.

A good example of this are oarfishes (*Regalecus*). Individuals of *Regalecus* spp. represent the three data points clustering very far away from the main regression line in Figure 7. It is unsurprising that *Regalecus* spp. represent such an extreme outlier to the remainder of the data, given this taxon has long been used as an example of the most extreme body proportions seen in fishes. However, in the context of the present study it is more notable that *Regalecus* spp. is more or less the only examined taxon that represents such an outlier. Other taxa, including other anguilliform fishes, show a very tight correlation between head and body proportions. This would not be expected unless there were some underlying physical and/or developmental constraint keeping head and body proportions consistent in fishes. If head and body proportions did not strongly covary in fishes it would be expected that many taxa would show proportions like *Regalecus*.

Similarly, many surgeonfishes (Acanthuriformes) and members of the triggerfish/filefish clade (Balistoidei) show a distinctive skull shape in which the neurocranium and orbit are positioned very posteriorly on the overall head, with the skull seemingly "slung forward" and the gills and opercular series positioned ventral to the overall cranium. Indeed, examining skeletons of these groups shows that the opercular series and gills actually extend substantially anterior to the neurocranium. Because the orbit is positioned so far posteriorly on the overall head, this results in the distance between the orbit and the posterior margin of the branchial cavity to be very short, and thus results in a gross underestimate of body length in these taxa. There does appear to be a correlation between head and body proportions in Acanthuriformes and Balistoidei as in other fishes, but the unique anatomy of these fishes means the landmarks normally used to define head proportions in this study cannot be applied to these taxa.

Experimenting with images of a skeletonized specimen of *Balistoides virescens* [140] finds that if OOL is measured using the anterior end of the gill chamber as the anterior landmark total length in this taxon is predicted with very low error (See Supplementary File S3: Table S7). Although *Dunkleosteus* and other arthrodires also exhibit gills that are anteroventrally

positioned relative to the braincase, the same issue is unlikely to characterize arthrodires because the error in *Balistoides* and other taxa primarily stems from the posterior location of the orbit, rather than the ventral position of the gills. This is again demonstrated with the experiment with *Balistoides*, in which the total anteroposterior length of the neurocranial/branchial region correlates well with the length of the fish, despite the gills being ventral to the neurocranium rather than posterior. This is further supported by the fact that OOL generally produces accurate predictions of body length in smaller arthrodires (see below) and does not dramatically underestimate body size as would be expected if an evolutionary anteroventral shift of the gill chamber was not reflected in body length.

Macruriform taxa such as chimaeroids are noteworthy in that they simultaneously support and do not support the relationship seen here. On the one hand, OOL substantially underestimates total length in chimaeroids. On the other hand, the body shape of chimaeroids shows these animals mostly conform to the observation that head and body proportions usually mirror each other in fishes. Chimaeroids have a short head, a relatively short, rotund thorax (treated as the distance between the pectoral and pelvic fins), and much of the discrepancy in estimated length is driven by their elongate, whip-like tails (i.e., a macruriform body plan). If chimaeroids had a typical fusiform body plan with a shorter, forked heterocercal or homocercal tail, the model would likely predict their body length with higher accuracy. This, in turn, suggests that the present model may not be useful for predicting total length in ptyctodont placoderms, which show similar body proportions to extant chimaeroids [59,60].

Some of the most consistent outliers in the dataset, particularly among generalized fusiform to elongate fishes, are groupers (Serranidae), which show much longer OOLs than expected. This relationship is present even after excluding the length of the opercular flap, with OOL typically overestimating the length of groupers by nearly 35%. Membership in Serranidae is a significant variable when included in the model, regardless of whether this is the only covariate added (t = −24.82, $p < 0.001$; Supplementary File S3: Section S10.1.2), or if body shape categories are also considered (−24.37, $p < 0.001$). This does not appear to be due to a shift in the position of the orbit or gills, as in Balistoidei. Instead, serranids simply appear to have disproportionately large heads compared to other fishes, and this can be seen when comparing serranids to other fishes using OOL or overall head length [33]. Slopes between OOL and total length do not differ between serranids and other fusiform fishes (t = 0.133, $p = 0.894$; Supplementary File S3: Section S10.1.2.), indicating that while serranids have much larger heads than other fishes, they show a similar allometric relationship between OOL and body size.

A disproportionately large head appears to be a derived trait of groupers relative to other fishes. More specifically, this state appears to have evolved within Serranidae and characterizes the clade formed by *Cephalopholis*, *Epinephelus*, and their relatives. By contrast, basal serranids such as *Variola* and *Plectropomus* [141–143] show proportions more similar to other fishes. For this reason, groupers were either dropped from the regression model or membership in Serranidae was treated as an additional variable in subsetted analyses (see Supplementary File S3 for more details). The same is true of Holocentridae, which also show systematically higher residuals than other fusiform fishes, possibly due to their large orbits.

Finally, when discussing patterns in the present dataset, it is also worth mentioning which fishes show the lowest errors and highest accuracy rates when estimating total length via OOL: phylogenetically basal fishes with relatively unspecialized, fusiform body plans. These include squalid and carcharhiniform sharks, tarpons (Megalopidae), salmonids, clupeiforms, osteoglossiforms, and coelacanths (Supplementary File S3: Figure S6). Thus, it appears as though the model accurately predicts body length in generalized fishes but is less accurate in taxa with specialized body shapes. This improves confidence that OOL should accurately estimate total length in *Dunkleosteus* and other arthrodires, as arthrodires are also thought to exhibit generalized, fusiform body plans with proportions close to the ancestral pattern for all gnathostomes.

### 3.1.3. Effects of Snout Length

Although the present model focuses primarily on OOL, snout length does contribute to total body length in fishes and in the previous analyses caused errors when estimating the length of fishes with very elongate snouts (e.g., Lepisosteiformes, Belonidae). Therefore, a model was created including snout length as an additional variable in the hopes that describing the proportions of the head using different allometric relationships for major regions would improve the accuracy of body length predictions in arthrodires. The hypothesis in this case being OOL would explain the majority of variation in total length minus rostrum length and variation in rostral length could be described as an additional parameter.

Adding snout length to the model resulted in a statistically detectable effect (t = 34.07, $p < 0.001$) and produced substantially better values of $\%PE_{cf}$ (14.6% versus 17.6%), AIC, and BIC compared to the OOL model without snout length (Supplementary File S3: Section S12.1). However, when applied to arthrodires of known length, this model resulted in systematically smaller lengths for arthrodires. In some cases this resulted in lower error rates, but in others including snout length resulted in systematic and sometimes substantial underestimates of actual body length (Supplementary File S3: Table S12). Examination of the data finds arthrodires have shorter snouts relative to their body size than other fish clades. This is true whether snout length is measured relative to head length (t = −7.325, $p < 0.001$; Supplementary File S3: Figure S12 and Section S12.5) or relative to total length (t = −3.836, $p < 0.001$; Supplementary File S3: Figure S12 and Section S12.5). This may be due to the fact that most bony fishes have prognathic mouthparts that extend anterior to the neurocranium (Figure 8), whereas arthrodires either have subterminal mouths (e.g., *Coccosteus*; Figure 8B) or mouthparts that extend to the anterior level of the neurocranium (e.g., *Dunkleosteus*, see Figure 1D).

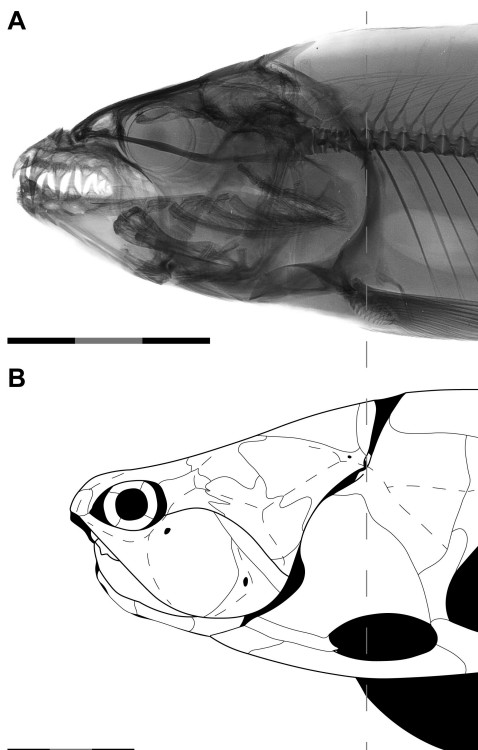

**Figure 8.** X-ray of *Hydrocynus forskahlii* ((**A**), CAS SU 63349; [144]) and *Coccosteus cuspidatus* ((**B**), modified from [15]) scaled to the same neurocranial length, showing how arthrodires have proportionally shorter snouts than other fishes. Note how the neurocranium of the two taxa is similar in shape but the head of *H. forskahlii* is slightly longer due to the more prognathic mouthparts of this taxon. Scale = 3 cm.

When snout length is included as an additional covariate estimated length for *Dunkleosteus* was substantially shorter than in other models (e.g., 2.72 m versus 3.53 m in CMNH 5768). This length is short to the point of being anatomically unjustifiable. Specifically, if CMNH 5768 were only 2.72 m in total length, the thorax would be too short to curve around the known dimensions of the thoracic armor and still end in a caudal fin and caudal peduncle, even assuming a proportionally deep peduncle as in *Coccosteus*, *Amazichthys*, serranids, or coelacanths. Adding snout length to the model but adding an interaction between snout length and clade (Supplementary File S3: Table S12), which in theory should compensate for the proportionally shorter snouts of arthrodires, still produces unrealistically low body length estimates (2.78–2.89 m) and 95% prediction intervals that are even larger than just considering OOL alone. This is because there are so few arthrodires known from complete remains the uncertainty in the interaction between snout length and clade effectively cancels out the improvement in accuracy from adding snout length as a variable. Thus, although including snout length improves model accuracy across fishes in general, it cannot be used when estimating length in arthrodires due to differences in cranial proportions between arthrodires and other gnathostomes.

3.1.4. Body Size of Arthrodires

OOL accurately predicts total length for arthrodires for which body fossils are known (Table 2, see also Supplementary File S3: Section S8). Without any additional qualifiers, OOL accurately predicts total length within ±12.5% in arthrodires (Supplementary File S3: Table S8). This is slightly better accuracy than the %PE for the model based on all fishes, and is largely due to the fact that almost all of the arthrodires examined in this study have a generalized fusiform body shape (i.e., there are no currently known anguilliform or discoid arthrodires). Using the model based on only fusiform fishes reduces this error to roughly 8–10% (Supplementary File S3: Table S10).

**Table 2.** Length estimates for arthrodires known from whole-body fossils using OOL from the all taxa, individual specimen equation. Selected representatives for each taxon are given in cases where more than one individual was measured for the sake of space, a complete listing of all results can be found in Supplementary File S3: Section S8. Abbreviations: PE, percent error; P.I., prediction interval. All measurements in cm.

| Taxon | Specimen | Actual Length | Estimated Length | +/−PE | 95% P.I. | PE |
|---|---|---|---|---|---|---|
| *Millerosteus minor* | FMNH PF 1089 | 13.7 | 13.87 | (11.4–16.3) | (8.9–21.6) | 1.1 |
| *Millerosteus minor* | Composite (see Methods) | 15.0 | 16.04 | (13.2–18.9) | (10.3–24.9) | 6.8 |
| *Africanaspis dorissa* | Reconstruction in [17] | 23.0 | 24.45 | (20.2–28.7) | (15.7–38.0) | 5.9 |
| *Incisoscutum ritchei* | Reconstruction in [55] | 30.3 | 31.62 | (26.1–37.2) | (20.4–49.1) | 4.3 |
| *Coccosteus cuspidatus* | NMS 1893.107.27 | 29.6 | 35.10 | (28.9–41.3) | (22.6–54.5) | 15.6 |
| *Coccosteus cuspidatus* | FMNH PF 1673 | 37.1 | 36.51 | (30.1–42.9) | (23.5–56.7) | −1.7 |
| *Coccosteus cuspidatus* | Reconstruction in [15] | 39.4 | 43.94 | (36.2–51.7) | (28.3–68.3) | 10.3 |
| *Coccosteus cuspidatus* | ROM VP 52664 | 37.5 | 42.52 | (35.1–50.0) | (27.4–66.1) | 11.8 |
| *Plourdosteus canadensis* | MNHM 2-177 | 37.5 | 51.40 | (42.4–60.4) | (33.1–79.9) | 27.0 |
| *Dickosteus threiplandi* | NMS 1987.7.118 | 43.7 | 56.13 | (46.3–66.0) | (36.1–87.2) | 22.2 |
| *Holonema westolii* | Reconstruction in [16] | 60.6 | 51.18 | (42.2–60.2) | (32.9–79.5) | −18.5 |
| *Watsonosteus fletti* | NMS G.1995.4.2 | 56.6 | 65.30 | (53.8–76.8) | (42.0–101.5) | 13.3 |
| *Amazichthys trinajsticae* | AA.MEM.DS.8 | 89.7 | 78.02 | (64.3–91.7) | (50.2–121.2) | −15.0 |

When examining coccosteomorphs, which are more conservative in body shape compared to Arthrodira as a whole, OOL shows slight positive allometry relative to body size (log slope = 1.14 ± 0.06; Supplementary File S3: Section S8.2). However, when considering all arthrodires, allometry of OOL is near isometric (log slope = 0.97 ± 0.07). The differences between these two methods appear to be driven by *Amazichthys*, which has an unusually small head relative to the line formed by all other arthrodires (Supplementary File S3:

Figure S17) Given the small sample size of arthrodires here (N = 17, 13 of which are coccosteomorphs), future studies may show stronger evidence for interspecific isometry or allometry in head size across Arthrodira. However, the fact that neither analysis shows significant negative allometry (i.e., proportionally smaller heads at larger sizes) suggests that OOL is unlikely to substantially underestimate body size in *Dunkleosteus terrelli*.

Coccosteomorphs tend to show slight overestimates of total length. This may be due to the fact that in many coccosteomorphs the cranio–thoracic joint overhangs the body (Figure 1C), and thus the neurocranium and branchial region are slightly farther forward than would be expected if measuring OOL to the cranio–thoracic joint. Another possibility is ecology: extant fishes with similar ecological habits to those proposed for *Coccosteus*, *Watsonosteus*, *Dickosteus*, and *Plourdosteus* (benthic/demersal freshwater or estuarine piscivores, such as Channidae) tend to have slightly longer OOL than expected. Alternatively, this error might be due to flattening of the specimens distorting OOL, resulting in OOL being measured as the tangential distance between the two points (the cranio–thoracic joint is slightly dorsal to the orbital margin) rather than the natural OOL along the anteroposterior axis. This might be supported by the fact that the reconstruction of *Coccosteus cuspidatus* in Miles and Westoll [15] more closely conforms to theoretical expectations than many of the actual fossils.

Finally, these overestimates may be because in several coccosteomorph fossils examined it is difficult to determine whether the tail is preserved all the way to its tip. For example, the *Coccosteus* fossils examined here seem to have a shorter post-thoracic region than the reconstruction in Miles and Westoll [15], and it is not clear if this is because the tip of the tail is not preserved in these specimens or if the reconstruction in Miles and Westoll [15] has a post-thoracic region that is too long. In arthrodires from the Achanarras beds of Scotland (i.e., *Millerosteus*, *Coccosteus*, *Dickosteus*, and *Watsonosteus*), specimens in museum collections are often unprepared after initial collection (M. Newman, pers. comm.). This results in more subtle details of the skeleton (in particular the extent of the caudal fin, and thus total length) being difficult to determine. Fully prepared specimens in private collections from the same localities often display spectacular preservation to the tip of the caudal fin (M. Newman and R. Jones, pers. comm.). This is supported by the fact that specimens of *Coccosteus* and *Millerosteus* from the FMNH and ROM, which have undergone more preparation, have longer bodies and proportions that correlate more closely with theoretical expectations. It is possible the Achannaras arthrodires might be slightly longer than reported here due to incomplete preparation, but at the same time it is also clear arthrodires have a shorter, squatter torso than extant eugnathostomes, with the post-thoracic region being particularly short.

In contrast to coccosteomorphs, OOL tends to underestimate length in the aspinothoracidan *Amazichthys trinajsticae* and the basal arthrodire *Holonema westolii*. However, *Amazichthys*, at the very least, shows a body plan that suggests some degree of axial elongation has occurred, similar to extant mackerels (see "Body Shape of Arthrodires", below). No complete body fossils are known for dunkleosteoids like *Dunkleosteus*, but given the phylogenetic position and anatomy of *Dunkleosteus* the head–body proportions of this taxon might be expected to be intermediate between aspinothoracidans and coccosteomorphs (i.e., *D. terrelli* neither shows signs of axial elongation in its armor proportions nor the overhanging head of coccosteomorphs).

None of this detracts from the main point that OOL accurately predicts total length in arthrodires, and it generally does so with higher accuracy than for fishes as a whole. Therefore, this suggests that OOL should accurately predict total length in arthrodires for which post-thoracic remains are unknown such as *Dunkleosteus terrelli*.

### 3.2. Body Size of Dunkleosteus terrelli

3.2.1. Length of *Dunkleosteus terrelli*

For this section, the results of the OOL model will primarily focus on CMNH 5768 (Figure 5). This is the largest mounted individual of *Dunkleosteus terrelli*, as well as the

specimen that will be most familiar to readers as it serves as the basis for the majority of *Dunkleosteus* casts seen throughout the world. CMNH 5768 is one of the largest known individuals of *D. terrelli*, having the third largest inferognathal out of 68 specimens examined in the collections of the CMNH, AMNH, and NHMUK, and thus is within the upper 5th percentile for body size in this species. Reliable identifiers of ontogenetic maturity have yet to be identified in arthrodires (though see [145]), but preliminary observations of the *Dunkleosteus* hypodigm and unpublished comparisons by the present author with ontogenetic size distributions of similar living fishes (e.g., pelagic sharks) suggests CMNH 5768 represents a large, sexually mature adult. Thus, CMNH 5768 can be considered a "representative" adult *D. terrelli* and a useful point of comparison when talking about the average adult size of this species under different models. Length estimates were produced for other, smaller specimens (see Methods), but because these specimens are juveniles or young adults they provide little context as to the present question of the adult size of *Dunkleosteus*. However, these length estimates are included in Supplementary File S3 for comparison.

Estimating the length of CMNH 5768 using OOL and no additional qualifiers produces a total length of 352.6 cm (+/−%PE, 290.7–414.5 cm) using individual measurements and 338.9 cm (+/−%PE, 278.4–399.4 cm) (Table 3) using species averages. This prediction can be refined even further. The highest prediction errors occur in fishes with highly specialized body shapes, namely extremely short discoid/compressiform fishes or extremely elongate/anguilliform ones. It is highly unlikely that *Dunkleosteus* had a compressiform or anguilliform body plan. All eubrachythoracid arthrodires for which the body shape is known (e.g., *Coccosteus*, *Millerosteus*, *Watsonosteus*, *Incisoscutum*, *Plourdosteus*, *Torosteus*) show relatively generalized fusiform body plans [15,20,146], with the exception of the benthic heterosteiids [23] which show a flattened (but not necessarily short or elongate) shape. Even *Amazichthys* exhibits a somewhat fusiform shape despite its elongate trunk. It is possible that some later arthrodires (specifically selenosteid aspinothoracidans), may have experimented with alternative body shapes, especially given the great morphological disparity seen in aspinothoracidans [21,147–149], but there is currently no evidence that *Dunkleosteus* and other non-heterosteiid dunkleosteoids deviated from a fusiform body plan. The proportions of the thoracic armor do suggest that *Dunkleosteus* had a relatively deep body relative to its length (see below). However, the thoracic armor suggests that *Dunkleosteus* was deep-bodied but still fusiform in the manner of a tuna or lamniform shark, rather than truly compressiform/discoid. Thus, anguilliform and compressiform fishes can be safely excluded from the dataset to improve estimation accuracy.

Removing fishes with highly specialized body shapes results in very little change to estimated lengths; 353.8 cm (±%PE: 299.8–407.8 cm) for individual specimens and 343.0 cm (±%PE: 278.4–399.4 cm) for species-averages. The relative lack of change relative to the all species model appears to be due to an over-representation of elongate bodied istiophoriforms at large sizes. The over-representation by these fishes also appears to be why some of the models looking at individual specimens (specifically, the all fishes, fusiform + elongate fishes, and pelagic taxa only models; Table 3) produce lengths that are slightly greater than all other models (~350 versus 310–340 cm). Going to an even greater extreme, and estimating length in *Dunkleosteus* using only fusiform fishes (i.e., excluding fishes with elongate body plans like *Sphyraena* or *Tetrapturus*) produces an estimated body length of 319.7 cm (±%PE: 281.7–357.6 cm) using individual specimens and 313.9 cm (±%PE: 278.6–349.3 cm) using species averages. Similarly, adding shape as a categorical covariate produces lengths of 324.8 cm (±%PE: 285.8–363.9 cm) for individual specimens and 320.1 cm (±%PE: 279.8–360.3 cm) for species averages.

**Table 3.** Length estimates of the largest complete individual of *Dunkleosteus terrelli* (CMNH 5768) under a variety of different models and starting assumptions. Abbreviations as in Table 2. All measurements in cm.

| Model | Individual Specimens | | | Species Averages | | |
|---|---|---|---|---|---|---|
| | Estimate | +/−PE | 95% P.I. | Estimate | +/−PE | 95% P.I. |
| All fishes | 352.6 | (290.7–414.5) | (226.8–548.1) | 338.9 | (278.4–399.4) | (214.0–536.7) |
| Fusiform and elongate fishes | 353.8 | (299.8–407.8) | (241.7–518.0) | 343.0 | (289.9–396.1) | (229.7–512.1) |
| With shape as covariate | 324.8 | (285.8–363.9) | (237.8–443.7) | 320.1 | (279.8–360.3) | (229.9–445.6) |
| Fusiform taxa only | 319.7 | (281.7–357.6) | (236.1–432.8) | 313.9 | (278.6–349.3) | (234.1–421.1) |
| With body depth as covariate | 335.4 | (281.1–389.6) | (221.4–508.0) | 344.1 | (283.6–404.6) | (221.8–536.6) |
| Including snout length as a separate integer | 336.8 | (284.9–388.7) | (231.1–492.7) | 328.5 | (276.1–380.9) | (219.8–493.2) |
| Pelagic taxa | 357.5 | (298.3–416.7) | (242.4–527.2) | 328.8 | (276.7–380.9) | (222.4–486.1) |
| Fusiform and elongate non-acanthopterygians | 340.5 | (285.1–395.9) | (225.7–513.7) | 318.5 | (279.3–357.7) | (234.0–433.5) |
| Sharks | 298.5 | (264.2–332.9) | (224.1–397.8) | 299.6 | (268.0–331.2) | (227.9–393.9) |
| With shape and variable slope for Chondrichthyes | 340.7 | (298.4–382.9) | (245.1–473.6) | 328.6 | (284.4–372.9) | (226.9–476.0) |
| Head length | 266.7 | (228.3–305.0) | (184.2–386.0) | 262.3 | (221.3–303.2) | (176.0–390.9) |
| **Other methods of estimating length** | | | | | | |
| Scaling from *Coccosteus* in [15], head length | | | | 341 | — | — |
| Scaling from *Coccosteus* in [15], length of mediodorsal (sensu [64]) | | | | 223 | — | — |
| Scaling from *Coccosteus* in [15], greatest external length of mediodorsal | | | | 297 | — | — |
| Scaling from *Coccosteus* in [15], greatest length of posteroventrolateral | | | | 388 | — | — |
| Scaling from *Coccosteus* in [15], inferognathal length | | | | 523 | — | — |
| Scaling from *Coccosteus* in [15], body depth | | | | 614 | — | — |
| Entering angle (sensu [150]) | | | | 347 | — | — |
| Approximate location of pelvic girdle on body | | | | ~340 | — | — |

Another potential factor that might be useable to constrain body size in *Dunkleosteus* is inferred life habits. Some groups of marine vertebrates, such as sharks, thunnins, and ichthyosaurs, show higher aspect ratios and shorter, deeper bodies compared to neritic or demersal members of the same group [12,151,152]. However, Friedman et al. [153] found the opposite pattern: pelagic fishes show narrower bodies and lower aspect ratios than demersal fishes. Despite this disagreement in the specific patten the implication is clear: life habits can have a significant influence on body shape (and therefore estimated length) in fishes. Given *Dunkleosteus* is also thought to have been an open-water fish [11], it is possible this taxon might exhibit a body plan more similar to other pelagic fishes than demersal, benthic, or neritic fishes. Using a model containing only pelagic taxa produces a length of 357.5 cm for CMNH 5768 using model based on individual specimens, and 328.8 cm for a model based on species averages (Table 3). Pelagic fishes in general have slightly longer bodies relative to their head than demersal fishes (Supplementary File S3: Section S10.2.2), though this pattern is not observed in pelagic chondrichthyans or most scombrids outside of Scomberomorini.

Arthrodires are frequently reconstructed based on analogy with modern sharks (e.g., [11,12,57]). This is based on similarities in the inferred morphology of the two groups (cartilaginous endoskeleton [61], both groups being "dismemberment" predators [33], presence of claspers and viviparity [58,154]), and also on the (not always accurate; see [155,156]) idea that sharks are the best extant models for the early gnathostome condition. Thus, it is worth considering whether a shark-only model performs substantially different from one based on all fishes (i.e., one including data from acanthopterygians and other derived teleosts). A regression model using only sharks produces much shorter lengths than most other models (298.5 cm versus 310–350 cm; Table 3). However, this is largely because most of the very largest sharks are short-bodied pelagic sharks such as lamnids, which causes the regression line to slope downward at larger body sizes (see above). However, at the same

time the body proportions of *Dunkleosteus* would be expected to be more similar to a lamnid (i.e., shorter abdomen, longer head) than any other extant fish, based on inferred similarities in paleoecology between the two taxa [11,12]. Despite this, it is noteworthy that the 95% prediction intervals for the shark-only model, despite likely being unreasonably wide (see below), fail to even exceed 4.0 m. Similarly, considering only elongate-bodied or fusiform non-acanthopterygian fishes except acanthopterygian fishes produced a total length of 340.5 cm (Table 3), not greatly different from models including acanthopterygian taxa.

Another observation that might be useful to help constrain body length estimates in *Dunkleosteus* is the dimensions of the thoracic armor. The body of *Dunkleosteus* can only be so short relative to the length of the animal in order to accommodate the armor's great depth and width, as well as the fact the body must be long enough to allow room for a pelvic girdle and fins, claspers (which form a separate limb girdle in arthrodires; see [154]), anal plate (and possible anal fin), and caudal fin. Going further, if the estimated length is too short it violates the known proportions of the specimen represented by the ventral armor. CMNH 5768 is approximately 137.5 cm long from the anterior tip of the rostral bone to the posterior tip of the postroventrolateral plates on the ventral shield, so the animal clearly cannot be shorter than this.

Lengths of less than 3.05 m for CMNH 5768 are unlikely because beyond this point the body would have to curve much more than can be anatomically justified between the deepest and widest parts of the body (based on the dimensions of the thoracic and ventral armor) and the caudal peduncle, even accounting for retrodeformation of the ventral shield (Supplementary File S5). Additionally, at lengths shorter than 3.05 m there is very little room on the reconstructed animal for a pelvic girdle or anal fin (with most of the post-armor length being occupied by the estimated length of the caudal fin), and the body has effectively lost its fusiform shape, resembling a highly discoid opah (*Lampris*) or pacu (*Colossoma*). This seems highly unlikely from a hydrodynamic perspective, especially as *Dunkleosteus* lacks the mediolaterally narrow body seen in most actinopterygians and the resulting animal would not be discoid (and at least somewhat hydrodynamic), but a swimming sphere. Thus, models that produce much shorter body length estimates, such as the shark-only model, seem unlikely.

When adding body height as an additional variable the model produces lengths that are very similar to those considering only OOL (323.7–342.6 cm depending on how body depth is accounted for; Supplementary File S3: Section S11). Of these models, the model that treated relative body depth as a ratio between head length and body depth showed the greatest improvement in AIC and the lowest error values, and produced a length of 335.3 cm (see Table 3). Thus, the relatively deep thorax of *Dunkleosteus* does not appear to suggest a particularly larger animal than estimated via OOL.

Because arthrodires seem to have a significantly shorter snout than the average fish (see "Effects of Snout Length, above), and accounting for this reduces differences in head–body proportions between arthrodiran and non-arthrodiran fishes (Supplementary File S3: Figure S5), a model was fit using OOL to estimate total length minus snout length, and then adding snout length back on as a known value. This produced lengths of 336.8 cm for CMNH 5768 using individual specimens and 328.5 cm under species averages (Table 3). Errors across fishes in general using this method were slightly lower than using OOL to measure total length as a whole (14.2% versus 17.6%).

Finally, it is possible to consider body shape and clade membership together (i.e., with interactive effects) to try and improve prediction accuracy in *Dunkleosteus terrelli*. This method produced lengths of only 271.7 m for CMNH 5768 under individual data points and 294.2 under species averages (Supplementary File S3: Table S10), too short to be realistic given the dimensions of the armor. Additionally, the prediction intervals for these models were very wide. Much of this is due to the fact that there are very few arthrodires for which total length is known, and all of these are ≤1 m in length. These taxa are required to determine the coefficient of the allometric slope for Arthrodira. Thus, the unrealistically short lengths produced for *D. terrelli* here are likely the result of extrapolation error [30],

which is supported by the fact that these models accurately estimate total length in smaller arthrodires (Supplementary File S3: Table S10).

Further examination found that the primary differences in regression slope were between chondrichthyans and all other fishes, and arthrodires did not have significantly different slopes from actinopterygians and sarcopterygians (Supplementary File S3: Section S10.7). This is likely driven by the non-random relationship between body shape and body size in elasmobranchs (see "Results of Model", above). Thus, it is possible to consider the difference in slope between Chondrichthyes and all other fishes to be the only relevant phylogenetic difference in allometry in order to improve statistical power. This is supported by statistical analysis (Supplementary File S3: Section S10.7.3). This method produces length of 340.7 cm (%PE: 298.4–382.9 cm) for CMNH 5768 under individual data points and 328.6 cm (%PE: 284.4–372.9 cm) under species averages (Table 3), very similar to the other methods.

Overall, the model considering shape differences and allowing for variable slopes between chondrichthyans and all other fishes is considered to be the best fitting model for the present data given it accounts for most of the potential biases identified in this study. It also produces some of the lowest error rates (%PE$_{cf}$ = 12.40) and lowest values of AIC ($-2310$) and BIC ($-2237$). The only model that produced comparable values was considering shape as the only additional variable (%PE$_{cf}$ = 12.03, AIC = $-2846$, BIC = $-2785$), but this model is not favored as it has issues when predicting length in large fishes. Specifically, because the largest fishes in this study are primarily lamnids, megachasmids, or echinorhinids which are fusiform but have very short trunks and long branchial regions (Supplementary File S3: Figure S10), this results in the model considering only body shape to be biased towards smaller lengths at larger body sizes. Thus, the latter model's inability to distinguish lamnids from other fusiform fishes makes it less ideal for estimating shape in *Dunkleosteus*.

The 95% prediction intervals for length estimates of *Dunkleosteus* in this study are extremely large, often ± 50% of estimated body length. However, this is a consequence of the log-transformation of the data and subsequent back-transformation, and is a common issue when predicting data on a logarithmic scale [40,101,114,157]. In effect, predicting data on a log-transformed scale means the prediction intervals are also log-distributed, and this results in small errors and small numbers of outliers being disproportionately magnified when back-transformed into arithmetic units. Specifically, back-transformation often transforms the residuals of the model from a normal distribution to a leptokurtic one (see, e.g., Supplementary File S3: Section S5.4.5). This stretches out the tails of the distribution and, because the residuals are used to calculate the prediction intervals, results in inappropriately wide prediction intervals for the detransformed data. Prediction and confidence intervals are very sensitive to departures from normality due to kurtosis, with potentially "catastrophic" [158] results if the data are very leptokurtic or platykurtic. Notably, this is an issue primarily applying to the prediction intervals of log-transformed models; the point estimates of these models are still reliable.

This is a very serious concern given the widespread usage of log-transformation in biological studies, as it basically means estimates of body size in extinct organisms frequently have prediction intervals that are so large as to be uninformative (i.e., on the order of magnitudes). This can be seen in the present study (e.g., Table 3), in that even using multiple variables highly correlated with total length to predict length only produces slight improvements in the width of the prediction interval. Similar issues occur in other studies. For example, Engelman [36] estimated body mass in extinct rodents using a multivariate model with three variables each highly correlated with body mass (skull length, occipital condyle width, and head–body length), and this only resulted in a slight reduction in the width of the prediction interval compared to a single-variable model. This is one reason why some authors (e.g., [159]) prefer using %PE or %SEE to model uncertainty for log-transformed models, as these are not influenced by the extremely long tails of the statistical distribution. However, this is merely a stopgap in the absence of a more rigorous,

statistically tested method, and the author lacks the mathematical expertise to propose a viable alternative.

Predicting length on a non-log-transformed model results in much narrower prediction intervals for *Dunkleosteus*, but because of uneven taxon distribution and model heteroskedasticity this model is statistically not supported over the log-transformed model. In fact, many of the 95% prediction intervals for smaller arthrodires in a non-log-transformed model allow for negative values of total length (Supplementary File S3: Table S5), a consequence of the heteroskedasticity in the dataset. That is, the magnitude of error in the correlation between OOL and total length is proportional to the size of the animal rather than uniform across the sample, requiring log-transformation. OOL may predict length in this dataset within ±17% of the actual value, but that imprecision may be 3.5 cm in a 20 cm-long minnow and 85 cm in a 5 m-long great white shark. Least-squares regressions are based on the assumption that the magnitude of errors is randomly distributed with respect to the regression line (i.e., homoscedastic, [160]), and failure to conform to this expectation can result in errors in line fitting and value estimation. This can be seen in in the non-log-transformed estimates, where the prediction interval is consistently ± 46.5 cm (Supplementary File S3: Table S5), regardless if the fish in question is a 40 cm *Coccosteus* or a 350 cm *Dunkleosteus*.

The upper 95% prediction intervals of most of the regression models technically allow for lengths of 5.0–5.5 m for CMNH 5768 (Table 3). However, there is reason to believe that the 95% prediction intervals are too conservative and that the actual possible range of body sizes for *Dunkleosteus* is much narrower. 88% of all fish species surveyed in the present study have estimated lengths that are within +/−33% of the actual value, and 67% have estimated lengths that are within ±20% of the actual value (Supplementary File S3: Section S5.7). Many of the fish species that fall outside this interval are those with unusual morphologies or have specialized body shapes, such as *Regalecus*, other anguilliform taxa, Balistoidei, Serranidae, or macruriform species. Indeed, the average PE across all fishes in these models is only 13–19% of the actual length. Similarly, lengths for *Dunkleosteus* outside of the interval of ±%PE result in biologically unlikely proportions (Supplementary File S5). This suggests the 95% prediction intervals here do not represent reliable estimates of the potential range of lengths of *D. terrelli* could span. This agrees with what is seen in complete arthrodires: the regression models generally predict total length in these taxa to within ±12.5% of the actual value (Supplementary File S3: Section S8), suggesting the estimated lengths for *D. terrelli* here approximate the actual value.

In summary, head dimensions (primarily OOL) predict a length of 3.2–3.5 m for typical adults of *Dunkleosteus terrelli* under a variety of models. Incorporating additional biological information about this species (pelagic habits, fusiform body shape) further reduces the range in body length estimates to approximately 3.3–3.4 m. The general agreement in these models despite different starting assumptions suggests the predicted length of ~3.4 m is close to the actual value. Prediction intervals are relatively large due to issues with making predictions on log-transformed scales (namely exponentiation of error), but producing lengths larger than 4 m in a large, adult individual like CMNH 5768 requires anatomically unlikely proportions that are not supported by the dimensions of the specimen. Some models, namely the model using total head length, the models incorporating snout length, the model based on chondrichthyans, and the model allowing variable slopes for different fish clades, produce even shorter lengths (<3.0 m) for adult *Dunkleosteus*. However, these estimates are dubious given the hard constraints of the known dimensions of the trunk armor.

### 3.2.2. The Largest Dunkleosteus

The largest currently known specimen of *Dunkleosteus terrelli* is CMNH 5936 (Figure 9). This specimen is a partial isolated left inferognathal, approximately 25% larger than CMNH 5768 in linear dimensions. The oral region of the inferognathal measures 34 cm in length along the anteroposterior axis of the jaw (though the bone retains a slight natural curvature),

with the entire inferognathal estimated to have originally measured 68 cm in length based on more complete specimens.

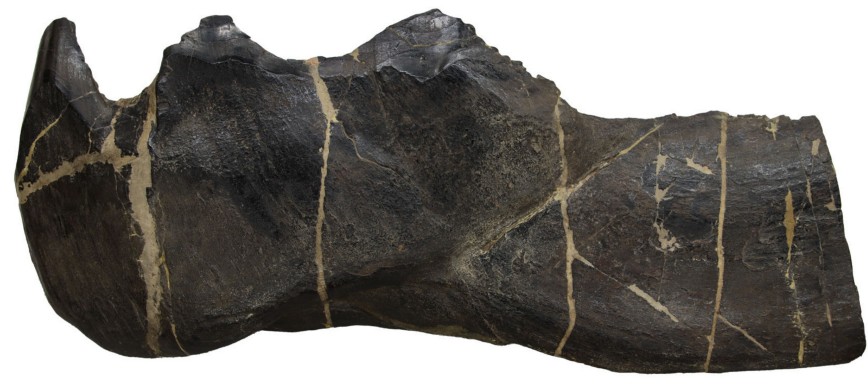

**Figure 9.** CMNH 5936, a left inferognathal fragment pertaining to the largest currently known individual of *Dunkleosteus terrelli*. Scale = 10 cm.

JM2, JM4, JM5, and total inferognathal length show slight positive allometry relative to OOL (and thus estimated total length) in *Dunkleosteus terrelli* (see Supplementary File S3: Table S14), with the lower jaw becoming larger and longer relative to body size throughout ontogeny. This can be observed firsthand when examining the mounted heads of *Dunkleosteus terrelli* at the CMNH, and agrees with previous studies regarding mouth size [33] and cheek plate dimensions [50] in *Dunkleosteus*. Given there is negative allometry in OOL in actinopterygians and lampreys, this could be interpreted as the mouth remaining isometric with regard to body size and OOL becoming proportionally smaller throughout ontogeny. However, there are three reasons this does not appear to be the case: (1) OOL scales isometrically in fishes that show direct development (see above), (2) positive allometry in mouth dimensions is visible in mounted specimens of *D. terrelli*, and (3) when scaling against head length, which is expected to scale isometrically in fishes, this pattern of positive allometry remains present (Supplementary File S3: Table S14) and in fact an additional measurement (JM3) also shows positive allometry.

Using the all-specimen model, the fusiform fishes only model, and the model including size as a categorical variable but allowing the slope for Chondrichthyes to differ from other fishes CMNH 5936, the largest known individual of *Dunkleosteus*, is estimated to measure between 339.4–423.5 cm in total length, with intervals for this overall range based on +/−%PE ranging from 293.7–497.8 cm (Table 4). The geometric mean of these estimates is 384.6 cm, with a median of 387.5 cm. The best-fitting models in this analysis, namely the species average, all specimens model and the two models with body shape and a variable slope for Chondrichthyes, produce slightly higher estimates, between 395.4–423.5 cm in length. Therefore, the current best estimates for CMNH 5936 suggest that it pertains to an individual of *D. terrelli* approximately ~410 cm in total length. Not only is this estimate much lower than the length of 8.79 m predicted for this individual by Ferrón et al. [12], but the very maximum limits of the uncertainty for this specimen using +/−PE (i.e., 4.5–5.0 m) are comparable to what previous studies have typically considered typical sizes for adult *D. terrelli* (e.g., [11]).

The non-gnathal material in the hypodigm of *D. terrelli* has not been surveyed as extensively but the overall size distribution is similar: there are several individuals similar in size to CMNH 5768, a few reach larger sizes comparable to CMNH 5936 (e.g., CMNH 9951), but no specimens significantly larger than CMNH 5936 have been identified. Thus, there are currently no individuals in the hypodigm of *D. terrelli* that could potentially pertain to an individual 4.7 m or greater in length, as would be necessary to produce the monstrous sizes reported for this species in previous studies. Indeed, larger (=likely adult) individuals of *Dunkleosteus* tend to be closer in size to CMNH 5768 than CMNH 5936.

Therefore, although the maximal length of *D. terrelli* is likely ~4.1 m, 3.4 m is a more typical adult size for this species.

**Table 4.** Estimated lengths (in cm) of the largest known specimen of *Dunkleosteus terrelli* (CMNH 5936) using the best-performing models in this study. Abbreviations as in Table 2. All measurements in cm.

| Measurement | Model | Data Type | Estimated Length | +/−PE | 95% P.I. |
|---|---|---|---|---|---|
| JM3 | All specimens | Individual Data | 409.4 | (337.6–481.3) | (263.4–636.5) |
| | All specimens | Species Averages | 392.7 | (322.6–462.8) | (248.0–622.0) |
| | Fusiform fishes only | Individual Data | 369.8 | (325.9–413.7) | (273.1–500.7) |
| | Fusiform fishes only | Species Averages | 362.7 | (321.9–403.6) | (270.4–486.7) |
| | Variable slope for chondrichthyans | Individual Data | 395.4 | (346.4–444.5) | (284.4–549.8) |
| | Variable slope for chondrichthyans | Species Averages | 339.4 | (293.7–385.1) | (179.0–643.6) |
| JM5 | All specimens | Individual Data | 423.5 | (349.2–497.8) | (272.4–658.4) |
| | All specimens | Species Averages | 406.0 | (333.6–478.5) | (256.4–643.1) |
| | Fusiform fishes only | Individual Data | 382.2 | (336.8–427.6) | (282.3–517.5) |
| | Fusiform fishes only | Species Averages | 374.8 | (332.6–417.0) | (279.4–502.9) |
| | Variable slope for chondrichthyans | Individual Data | 409.0 | (358.3–459.7) | (294.2–568.6) |
| | Variable slope for chondrichthyans | Species Averages | 350.5 | (303.3–397.7) | (183.7–668.8) |

### 3.2.3. Weight of *Dunkleosteus terrelli*

The modified ellipsoid model predicts body mass in fishes with a high degree of accuracy ($r^2$ = 0.992; %$PE_{cf}$ = 21.97). This is a much higher error than in length–weight models in other studies of fishes (e.g., ~10% PE; [161]), and much higher than the error of 1% reported by Ault and Luo [118]. However, this higher error is to be expected given this study is using an interspecific model with fishes of different body shapes, whereas most length–weight equations focus on a single taxon or a few closely related taxa of similar body shape. Another source of error is many fishes considered here had their weight estimated via standard length–weight models, which may not be sensitive to intraspecific variation in girth due to body condition. Given these limitations a model error of only 20% is rather good, especially as prediction errors for body mass in other vertebrate groups (e.g., mammals; [40,101]) are rarely below 33%. The model based on only large, pelagic fishes has a higher accuracy rate (%$PE_{cf}$ = 9.8) but a lower $r^2$ (0.990) due to the smaller range of body sizes in these data (see Materials and Methods).

The three main models considered here largely produce similar estimates of body mass for *Dunkleosteus terrelli* (Table 5). The model based on all fishes predicts a body mass of 106.7 kg for CMNH 7424, 391.7 kg for CMNH 6090, 381.4 kg for CMNH 7054, and 1008.4 kg for CMNH 5768 (Table 5). Assuming a swim bladder was present in *Dunkleosteus* results in a slight increase in mass of about 100 kg for CMNH 5768 (~1115 kg), though the presence of a swim bladder in arthrodires is dubious [57]. Estimated body mass for CMNH 5768 using only large, pelagic fishes (Lamniformes, Scombridae, and Istiophoriformes) results in a body mass of 1204.1 kg. Using the length–weight relationship of *Carcharodon carcharias* from Mollet and Cailliet [123] produces an estimated body mass of 941.3 kg for CMNH 5768 (Table 5; no prediction intervals available for this model). The 95% prediction intervals for the ellipsoid model are very large (e.g., 564.6–1801.0 kg for CMNH 5768), but this is attributable to the same factors that cause prediction intervals for length estimates to be unrealistically wide. Thus, based on the general agreement between these results, it seems reasonable to conclude that typical adult individuals of *D. terrelli* (i.e., the size of CMNH 5768) could reach weights of 950–1200 kg. More precise estimates might be obtainable via volumetric modeling [162], but for now the present estimates serve as a reasonable approximation of body mass in *Dunkleosteus terrelli*.

**Table 5.** Estimated body masses of *Dunkleosteus terrelli* and their 95% prediction intervals in kg. Prediction intervals not available for *Carcharodon* length–weight equation. Length calculated using the model including information from body shape and varying slope level for Chondrichthyans. CMNH 6090 and 7054 are almost identical in size, but the thoracic armor of 6090 is slightly deeper, hence the discrepancy in length and weight. Body proportions of CMNH 5396 were calculated assuming isometry with CMNH 5768. Additional details of how these masses were calculated can be found in Supplementary File S3.

| Specimen | Estimated Total Length (cm) | Ellipsoid Model, All Fishes | Ellipsoid Model, Large Pelagic Fishes | *Carcharodon* Length–Weight Equation |
|---|---|---|---|---|
| CMNH 7424 | 188.9 | 106.7 (60.5–188.4) | 166.7 (120.7–230.1) | 136.0 |
| CMNH 6090 | 283.2 | 391.7 (221.1–693.9) | 561.3 (401.8–784.2) | 423.9 |
| CMNH 7054 | 295.5 | 381.4 (215.5–675.0) | 545.0 (393.0–755.8) | 413.2 |
| CMNH 5768 | 340.6 | 1008.4 (564.6–1801.0) | 1204.1 (833.1–174053) | 941.5 |
| CMNH 5936 | 406.5 | 1763.9 (982.1–3168.0) | 1731.6 (1175.9–2549.8) | 1494.2 |

Overall, arthrodires appear to be much heavier relative to their length than modern sharks. This can be seen when plotting estimated body masses for Arthrodira against thunnins and sharks (Figure 10). Arthrodires are much heavier than sharks at similar lengths, regardless of their body size and the method used to estimate body mass. At smaller body sizes (i.e., *Coccosteus cuspidatus* or Gogo Formation arthrodires like *Compagopiscis croucheri*, *Incisoscutum ritchei*, and *Eastmanosteus calliaspis*) arthrodires show a length–weight relationship intermediate between thunnins and non-lamnid sharks, whereas the large, pelagic arthrodires of the Cleveland Shale (*Dunkleosteus terrelli*, *Bungartius perissus*, and *Heintzichthys gouldii*) show length–weight relationships comparable to extant tunas and lamnids. The weight for *Dunkleosteus* predicted here is significantly heavier than that of Carr [11], who predicted a weight of 665 kg for a 4.6 m *Dunkleosteus* assuming a length–weight relationship similar to extant sharks, despite much shorter length estimates for *D. terrelli* here. These results appear to be part of general differences in body shape between arthrodires, sharks, and other fishes, and further support the idea that sharks are not a good model for estimating the body proportions of arthrodires.

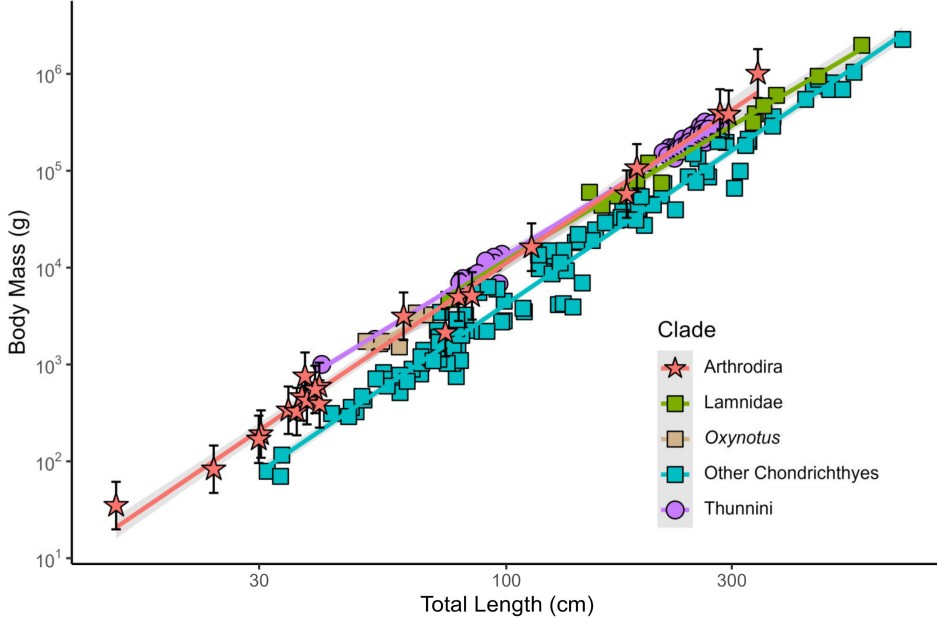

**Figure 10.** Graph of total length versus body mass for all specimens of sharks and tunas (Thunnini) in which weight was directly recorded, compared to estimated weights for arthrodires in this study.

Plotting a very crude length–weight curve for *Dunkleosteus* produces an allometric exponent of 3.56. This means that *Dunkleosteus* became more massive relative to its length as it grew. However, this allometric exponent is close to the general range seen in extant fishes (i.e., 2.5–3.5; [121]), if slightly higher. The thoracic armors of *Dunkleosteus terrelli* suggest this taxon became deeper-bodied throughout life [115], but it is also possible the present length–weight exponent could be biased by ontogeny. If *Dunkleosteus* showed a slight negative allometry between OOL and total length this would result in shorter lengths for juvenile individuals, and thus a slightly more isometric curve. Thus, although *Dunkleosteus* likely did show positive allometry between length and body mass based on multiple lines of evidence (calculated length–weight ratios, armor dimensions, etc.), the allometric exponent presented here might be a slight overestimate.

Notably, these methods of mass estimation make no attempt to account for the bony armor of *Dunkleosteus terrelli*. However, the armor plates of these animals may have less influence on body mass than might be assumed. The armor of one specimen of *D. terrelli*, CMNH 6090, has an estimated volume of 24.386 $cm^3$. Assuming an average density for whole bone (1.2–1.3 $g/cm^2$, [124,125]), this suggests the bony armor of this individual weighed ~30 kg, only 7.5% of the animal's predicted armor-free body mass (see Supplementary File S3: Section S16.6). These results are consistent even under different methods of estimating bone density (see Supplementary File S3: Table S16). This is much less than the carapace contributes to body mass in extant armored animals, such as nine-banded armadillos (*Dasypus novemcinctus*, 16% body mass; [163]) or turtles ($\geq$16.7% body mass; [164]).

Estimating the weight of the largest known specimen of *Dunkleosteus terrelli* (CMNH 5936), produces body masses ranging from 1494 kg to 1764 kg (Table 5). This possible range of masses seems likely given that when comparing the size of *D. terrelli* to a large individual of *Carcharodon carcharias* (e.g., MZL 23981, estimated weight ~2000 kg; see Figure 13 below), *Dunkleosteus* is much shorter in length despite having similar thoracic dimensions, and thus is expected to be less massive. Based on this, it seems the largest individuals of *Dunkleosteus terrelli* were smaller than the largest individuals of *Carcharodon carcharias*, the latter of which can attain weights of 2000–2500 kg [165–167] in large females.

## 4. Discussion

### 4.1. Head–Body Proportions in Fishes

The present study finds head and body proportions are very closely constrained in fishes, both in terms of the relationship between OOL and total length ($r^2$ = 0.95, %$PE_{cf}$ = 17.55; Table 1) as well as the aspect ratios of the head and body ($r^2$ = 0.80–0.88; Supplementary File S3: Table S3). Although other studies have noted that head length and total length are often correlated in fishes [168], the idea that these proportions are this strongly correlated across such a wide diversity of fishes is unexpected. Knapp et al. [169] find a similar correlation between head (= neurocranium) and body fineness ratio in Scombriformes, but the present study suggests that this pattern extends to all "fishes". The close relationship between head and trunk elongation in fishes (both in terms of total head length and OOL), used here to predict length in *Dunkleosteus*, occurs across such a great phylogenetic, morphological, and ecological breadth of fishes that it calls for a biological explanation. For example, one would expect a much poorer correlation between head and body proportions if a similar study was performed on tetrapods.

One possible developmental explanation for this pattern is it is caused by anteroposteriorly oriented morphogen gradients. In vertebrates, anteroposterior axis patterning is primarily governed by gradients of retinoic acid which diffuse from the anterior end of the embryo and fibroblast growth factors (FGFs)/WnT signaling proteins which diffuse from the posterior end [170,171]. Because these morphogens form opposing gradients in the developing embryo, their relative concentrations are used to signal the position of key anatomical boundaries such as those between somites [172,173]. More specifically, because an anteroposterior landmark is determined by relative concentrations of morphogens, its location would be proportional to the length of the embryo, and independent of the

embryo's length. This results in the head–trunk boundary being located at a consistent percentage of total length regardless of axial elongation.

Other results obtained here may be explainable via developmental mechanisms. Preorbital length may be more variable than head length or OOL because the jaws of vertebrates are formed by the first pharyngeal arch, which begins development in a ventral position but protrudes forward well after the head–trunk boundary is established [174]. Thus, preorbital length is less subject to control by morphogen gradients. Similarly, the non-linear correlation observed between fish head–trunk proportions may be caused by physical barriers to morphogen diffusion [175]. Even though the establishment of morphogen gradients is independent of embryo length, morphogens would still have to diffuse across a greater number of cell membranes in longer embryos, and this would slow the diffusion of morphogens. This would result in shorter-bodied fishes having larger heads relative to anteroposterior length and longer fishes having slightly shorter heads; the pattern observed here.

However, exactly how the location of the head–trunk boundary in vertebrates is determined [176] and how axial elongation effects the proportions of the developing embryo [46] appears to be poorly understood, making it difficult to offer a more comprehensive explanation. Specifically, the head–trunk boundary of vertebrates is significantly posterior to the unsegmented cranial mesoderm [176] and incorporates the anteriormost somites [177,178]. Thus, the mechanisms controlling the head–trunk boundary are not as simple as *hoxa1* and *hoxb1* defining the boundary between cranial and somitic mesoderm [179]. Expression of *hoxc6* may play a role, as this gene has been identified in specifying the position of the anteriormost thoracic vertebra in zebrafish [177] and tetrapods (birds, mice, and seemingly *Xenopus*; [180]). The consistent relationship between head–trunk proportions found here suggests a deeply conserved pattern potentially stretching back at least 450 million years, to the last common ancestor of Petromyzontiformes and Gnathostomata [49,181,182]. However, this hypothesis needs to be tested more extensively with the developmental biology of extant vertebrates and body proportions in Paleozoic jawless fishes.

To put this conclusion in less developmental biology-focused terms, when the anteroposterior axis is compressed or elongated in vertebrates, the head and body are usually compressed or elongated to similar degrees. Additionally, despite the great diversity of axial elongation in fishes [46,183], the location of the boundary between the head and the trunk as a percent of total length is remarkably conserved. Given the distribution of residuals in this study, variation in head-to-trunk proportions in fishes can best be described as random walk due to the phylogenetic accumulation of mutations allowing for slight shifts within an otherwise highly conserved pattern, with the possible exception of groupers (Serranidae) and some Lampriformes. However, a more in-depth analysis of these patterns (i.e., does this result follow a Brownian or Ornstein–Uhlenbeck model) is beyond the scope of the present study.

It is also very surprising that teleost and non-teleost fishes (e.g., sharks, basal actinopterygians, lampreys) show similar patterns of variation. If the head–trunk boundary is based on the position of cells along the anteroposterior axis of the embryo, one would expect homocercal teleosts to have different head–body proportions from heterocercal non-teleosts, due to the spinal cord extending to the tip of the caudal fin in the heterocercal taxa. However, this is not the case: teleosts and non-teleosts show similar head–trunk proportions despite differences in body shape (see "Body Shape of Arthrodires", below). It is tempting to attribute this pattern to biomechanical constraints (i.e., maintaining a hydrodynamic shape), but groupers (Serranidae) suggest otherwise, as they have highly deviant head-to-trunk proportions but maintain a fusiform shape.

The relationship between head dimensions (as OOL) and total length as recovered in this study can be best described as "OOL usually strongly correlates with total length, until it doesn't". OOL and total length usually correlates closely with one another in fishes, but in the few taxa where this relationship does not hold head/OOL–body proportions are often wildly different from other fishes. These unusual body proportions, in turn, are often associated with extreme specializations and life habits among extant fishes. The best exam-



ple of this are oarfishes (*Regalecus* spp.), which are massive outliers to every other species considered in this analysis (see outlying points in Figure 7). This conclusion is supported by the leptokurtic nature of the residuals [184]: most fishes cluster around a central mean value with a spread much narrower than expected for a normal distribution, but there are localized optima associated unusual, specialized body plans or morphologies (groupers, anguilliform taxa, macruriform taxa, certain piranhas like *Pygocentrus* spp., etc.). This suggests that head–body proportions in fish follow a punctuated equilibrium model [185], where head–body proportions are normally rigidly constrained under normal conditions by stabilizing selection. However, when fish move into adaptive zones where breaking this pattern provides more benefits than the costs imposed by stabilizing selection, selection proceeds at an extreme rate and often results in runaway selection. Thus, the distribution of head/OOL–body proportions in fishes is very narrow, but the outliers are much larger than would be predicted under a normal distribution.

Tetrapods seemingly show much more variation in head–body proportions than fishes (e.g., [186,187]), and this may be due to developmental fragmentation separating the head and trunk into distinct developmental modules [188–190]. This, in turn, is possibly linked with the loss of the branchial skeleton and evolution of a distinct neck during the transition to terrestrial life [191]. It is tempting to attribute the breaking of this constraint to one of the whole-genome duplications in "fish" evolution [192], but there is no evidence for a unique genome duplication event in tetrapods or sarcopterygians relative to other vertebrates [193–196]. Interestingly, Lampriformes (Actinopterygii) show some of the most extreme within-clade variation in head–trunk proportions among non-tetrapod vertebrates, ranging from the highly discoid *Lampris* (OOL overestimates total length by 20%) to the hyper-anguilliform *Regalecus* (OOL underestimates total length by 148%) (Supplementary File S3: Figure S6). It would be interesting to test if this largely mesopelagic fish clade converges with the primarily terrestrial Tetrapoda in the genetic/developmental mechanisms used to break the otherwise widespread constraint on head–body proportions in vertebrates.

It is clear that there is residual signal from body shape in the data, even after the broader pattern of allometric scaling is accounted for. That is, while the general mantra of "short fishes have short heads, long fishes have long heads" generally holds true (see Supplementary File S3: Figure S7), short fishes and elongate fishes have even shorter or more elongate bodies, respectively, than would be predicted based on the fineness ratio of the head alone. Discounting macruriform taxa, there is a clear pattern where the most negative residuals (overestimates of total length) are seen in compressiform taxa, the most positive residuals (underestimates of total length) are seen in anguilliform and elongate-bodied taxa, and fusiform fishes show residuals that cluster around zero (see Supplementary File S3: Figure S7). Additionally, within these groups, taxa that show body shapes intermediate between these broadly defined shape categories (e.g., "semi-compressiform" lamnids and thunnins or "semi-elongate" *Makaira* or *Mustelus*) show residuals that straddle the two categories. This variation is very subtle, less than $+/-3\%$ variation in OOL as a percentage of total length is enough to generate these patterns.

However, adding body depth to the model produced limited improvement in the accuracy of predictions. This suggests a potentially non-linear relationship between body aspect ratio and axial elongation (see also Supplementary File S3: Figure S11), with the effect of axial elongation accelerating at more extreme body shapes (this is particularly obvious in anguilliform taxa). This might be accomplished if different regions of the postcranium elongated at different rates relative to head elongation. The most obvious division of the axial skeleton that might show differential rates of axial elongation might be between the pre-anal and post-anal portions of the vertebral column, as this is considered the most significant division of the axial skeleton in fishes [74,197].

Analyzing these correlations between head elongation and aspect ratio in depth is far beyond the scope of the present study, with its focus on estimating body size in *Dunkleosteus*. Fortunately, this pattern is expected to have limited effect on length estimates

of *Dunkleosteus*, as fusiform fishes show little effect from this phenomenon and nearly all arthrodires are considered fusiform. It is possible that *Dunkleosteus* or some other arthrodires might have exhibited "semi-compressiform" body plans similar to modern lamnids or "semi-elongate" body plans similar to some istiophorids, but most arthrodires for which post-thoracic remains are known show generalized, fusiform body plans [15,19,20].

*4.2. Body Shape of Arthrodires*

The length estimates for *Dunkleosteus* presented here result in an animal with a very deep and wide body relative to its total length (Figure 11). This body shape appears to be real. Examining other arthrodires for which body proportions can be directly measured (e.g., *Coccosteus*, *Incisoscutum*) finds arthrodires in general have deep, wide, but relatively short bodies compared to other fishes (Supplementary File S3: Figure S7). However, *Dunkleosteus* appears to be extreme in this regard even among arthrodires. Fossils of *Dunkleosteus* consistently show proportionally deeper thoracic armors even compared to contemporary pelagic arthrodires like *Heintzichthys* and *Amazichthys* (Figure 11, see also [27]). Deep thoracic armors are present in all of the subadult to adult *Dunkleosteus* specimens examined in this study (CMNH 6090, 7054, and 5768), with the thoracic armor being deeper than the head is long. The thoracic armor of CMNH 5768 (the specimen that serves as the primary model for the present reconstruction) could have been slightly shallower (~10 cm) depending on how the armor is reconstructed, but the armor is complete in CMNH 6090 and 7054. The known dimensions of these fossils strongly limit how shallow-bodied any reconstruction of *Dunkleosteus* can be.

The shorter lengths predicted here force the body into a more lamnid-like configuration, as predicted by Ferrón et al. [12]. Otherwise, there is almost no way to reasonably fit a caudal fin within the limited post-thoracic length predicted by the model (~1.6 m) while still allowing the body to sufficiently curve to attach to a caudal peduncle (even a very deep one). The evolutionary and paleobiological implications of the deep body of *Dunkleosteus* are beyond the scope of the present study and are the subject of a manuscript in preparation by the present author [115]. However, it is worth noting that similar body shapes are actually common among pelagic vertebrates, including thunnin actinopterygians, pelagic lamniforms (compare Figure 13 to Figure 12C), and ichthyopterygian marine reptiles [198], among others. Such a body shape may actually be biomechanically expected among pelagic vertebrates, as it results in a lower surface-to-volume ratio and thus reduces drag while swimming [39].

Although the details of how the reconstruction presented here was made are the subject of a manuscript in preparation by the present author [115], two additional observations are worth detailing as they suggest the shorter lengths and deep body for *Dunkleosteus* presented in this study are real and not the result of statistical error. The first is the location of the pelvic girdle. In arthrodires for which the pelvic girdle is preserved in situ (including *Millerosteus*, *Coccosteus*, *Watsonosteus*, *Dickosteus*, *Incisoscutum*, *Plourdosteus*, *Amazichthys*, and *Heintzichthys*), the pelvic girdle is invariably located at or slightly posterior to the posterior end of the ventral shield [15,20,21,199] (Figure 11, Supplementary File S6). Most non-acanthomorph fishes have pelvic fins located roughly 45% (~35–50%) of the total length of the body, and roughly midway between the origin of the pectoral fin and base of the caudal fin (again, ~45% of this value; Supplementary File S3: Section S17.6). This pattern is present in sharks, sarcopterygians, non-acanthomorph actinopterygians (such as salmonids, catfishes, and minnows), and the few arthrodires in which the pelvic girdle is known (Supplementary File S6). Thus, the length of the ventral armor in arthrodires can be used to determine the position of the pelvic girdle, and the position of the pelvic girdle by extension can be used to approximate the length of the entire animal. Given the head and body armor of CMNH 5768 is only 1.38 m long, this would imply the entire animal was only ~3.4 m long (possible range ~3.0–3.9 m if making "guesstimate" allowances for interspecific variation in pelvic girdle location). This would result in a very deep body plan for *Dunkleosteus terrelli* due to the preserved depth of the thoracic armor relative to

estimated length. Lengths of 5+ m for CMNH 5768 would require a comparatively anterior location of the pelvic girdle (≤30% total length) and would result in body proportions more similar to acanthopterygian fishes, which are highly unlikely for *D. terrelli*.

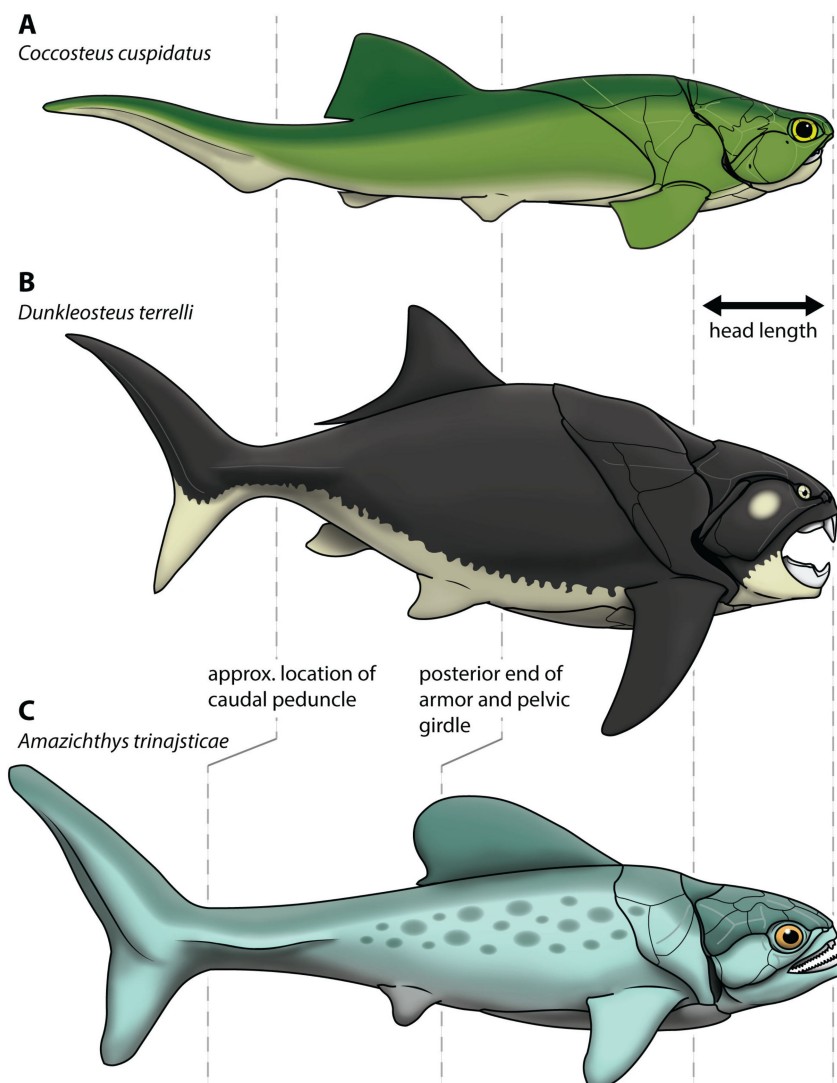

**Figure 11.** Reconstructions of (**A**) *Coccosteus cuspidatus* (modeled after [15,200]), (**B**) *Dunkleosteus terrelli*, and (**C**) *Amazichthys trinajsticae* (proportions modeled after [21]), scaled to the same head length. The shorter lengths for *Dunkleosteus terrelli* in the present study better agree with the locations of the pelvic girdle/posterior end of the ventral armor and caudal peduncle in other arthrodires. *A. trinajsticae* also shows a more elongate body plan than other arthrodires (especially if scaled based on OOL rather than head length, as here). Missing elements of *Amazichthys* modeled after *Draconichthys*, *Gymnotrachelus*, *Stenosteus*, and *Trachosteus*. Drawings by Russell Engelman.

The other observation that supports a shorter length for *Dunkleosteus terrelli* is the likely position of the anus. Trinajstic et al. [57] report the opening of the anus in *Incisoscutum* is located just posterior to the posteroventrolateral plates, where the pelvic girdle would have been. This agrees with the location of the anus relative to the claspers and pelvic girdle in extant chondrichthyans [201]. This is also the case in AMNH FF 2826 (referred to *Heintzichthys gouldii*), the only other arthrodire that preserves evidence of the location of the cloaca [26]. In this specimen, there are preserved gut contents (possibly the coprolitic infilling of a spiral intestine) that extend beneath the spinal column just posterior to the

ventral armor the until the gut contents reach the pelvic girdle, at which point they appears to breach the body wall. This suggests that the cloaca of *Heintzichthys* opened just posterior to the pelvic girdle (which, in turn, is located just posterior to the ventral armor), just as in *Incisoscutum* and modern chondrichthyans. The haemal arches of AMNH FF 2826 become much larger in size just posterior to the inferred location of the cloaca, suggesting this represents the posterior end of the visceral cavity, as in other fishes [197].

Assuming the anus of *Dunkleosteus* opened in a similar position to other arthrodires and extant chondrichthyans, this would mean in order for typical adults of *Dunkleosteus* (i.e., those the size of CMNH 5768) to be longer than 4 m the visceral cavity would have to be unusually small relative to the animal's size. Additionally, a significant portion of the animal's body would be composed of just the post-anal, precaudal region, unlike other arthrodires. While some extant fishes do exhibit this kind of morphology (i.e., *Electrophorus*, the electric eel) this arrangement only evolves in very specific evolutionary circumstances (i.e., the enlargement of the electric organ in *Electrophorus*, which occupies 80% of the animal's body length; [202]). Such an interpretation is unlikely for *Dunkleosteus terrelli* and the likely position of the anus supports smaller sizes for this species.

Overall, arthrodires appear to have body proportions unlike any fish alive today. Body shape diversity among generalized fishes examined in this study can largely be divided into three major groups: actinopterygians, elasmobranchs, and arthrodires (Figure 12). Each of these groups are highly distinctive in the relative proportions of their three major body axes (Table 6). This is especially apparent when scaling representatives of these three groups to the same body mass (Figure 12). Actinopterygians tend to have dorsoventrally deep and mediolaterally narrow bodies that are elliptical in cross-section (Figure 12A). Sharks, by contrast, have bodies that are much more anteroposteriorly elongate at the same mass (Figure 12C). Their bodies tend to be dorsoventrally shallow and mediolaterally narrow, but are also subcircular in cross-section with the diameter of the dorsoventral and mediolateral axes being close to equal (Supplementary File S3: Figure S7).

**Table 6.** Broad-scale differences in body proportions between the three major fish clades considered in this study. Note Chondrichthyes here almost exclusively refers to Elasmobranchii as extant holocephalians and batoids have heavily modified body plans (though extinct chondrichthyans like *Cladoselache* are generally similar in body shape to extant elasmobranchs), and Osteichthyes almost exclusively refers to actinopterygians due to low availability of data for sarcopterygians.

| Clade | Arthrodira | Chondrichthyes (Elasmobranchii) | Osteichthyes (Actinopterygii) |
|---|---|---|---|
| Body cross-section in anterior view | Circular | Circular | Mediolaterally narrow |
| Anteroposterior length relative to thoracic girth | Short | Elongate | Variable, generally intermediate |
| Body height relative to anteroposterior length | Deep | Shallow | Deep |

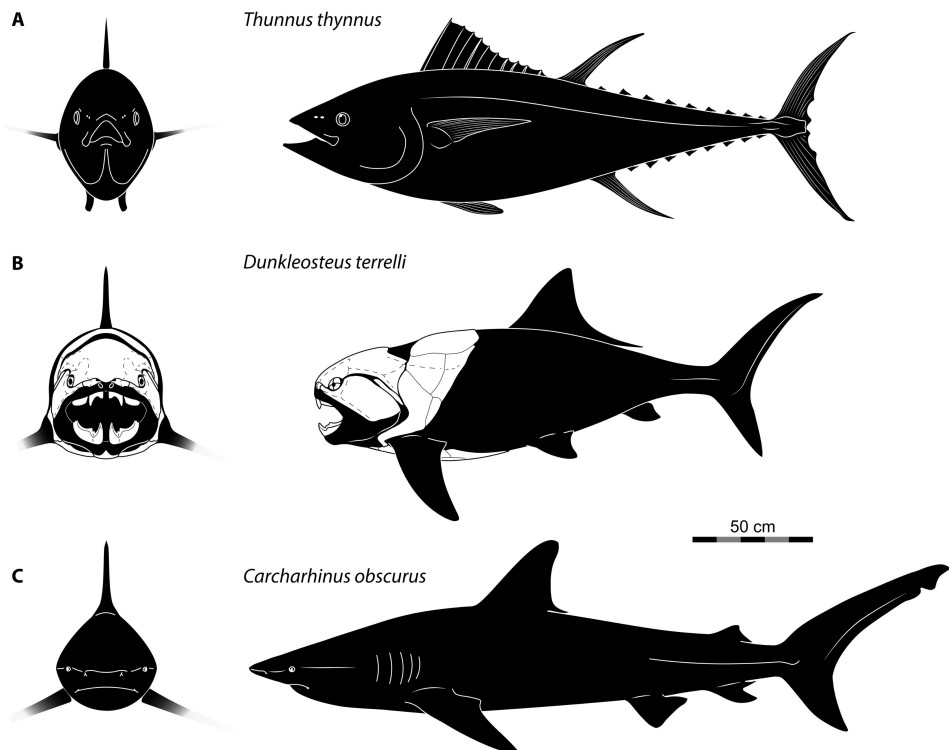

**Figure 12.** Silhouettes of a tuna ((**A**), *Thunnus thynnus*), *Dunkleosteus terrelli* (juvenile) (**B**), and carcharhinid shark ((**C**), *Carcharhinus obscurus*), all scaled to ~150 kg, showing differences in body shape of actinopterygians, arthrodires, and elasmobranchs at the same body mass. Proportions of (**A**) modeled after Rivas [203] and Russell [96]; (**B**) from present study and Engelman [115], based on CMNH 7424 and CMNH 6090; **C** modeled after 3.2 m individual in Garrick [37]. Note tunas are unusually wide for actinopterygians, the figured individual of *C. obscurus* is comparatively deep-bodied for *Carcharhinus*, and the length of *D. terrelli* might be a slight overestimate if arthrodires show ontogenetic allometry similar to actinopterygians. Thus, these three specimens understate the differences in body shape between the three major clades. Drawings by Russell Engelman.

Arthrodires resemble neither of these groups. Like actinopterygians, arthrodires tend to be deep-bodied, but like sharks are nearly subcircular in cross-section. This results in a body plan that is extremely girthy relative to its length, such that arthrodires have very short, squat bodies compared to both actinopterygians and elasmobranchs at the same body mass (Figure 12B). Indeed, arthrodires show higher estimated weights relative to their length than most sharks (Figure 10), more similar to tunas or lamnids. This occurs in all arthrodires, not just pelagic species like *Dunkleosteus*, also being present in demersal/neritic taxa such as *Coccosteus* and the Gogo arthrodires. Some arthrodires (mostly "selenosteid-grade" aspinothoracidans) deviate from this ancestrally short, deep body shape and show mediolaterally compressed (e.g., *Oxyosteus*, *Leptosteus*, and *Synauchenia*; [148]) or elongate (e.g., *Amazichthys*; [21]) morphologies. However, these are all secondary derivations of the ancestrally short, deep arthrodire body plan.

The only complete arthrodire that seems to lack a short, deep body is *Amazichthys trinajsticae* [21]. However, there is evidence that the body shape of this taxon is more complex than it appears. *Amazichthys* is characterized by both a small head relative to its body length and an unusually shallow body compared to other arthrodires [21], such that this taxon exhibits head–body proportions similar to living fishes with elongate trunks like *Coryphaena* (Engelman [33]: Figure 6). Thus, although *A. trinajsticae* superficially looks fusiform, it is elongate relative to the ancestral condition in arthrodires. Among extant fishes, the closest body shape analogue of *A. trinajsticae* would be a wahoo (*Acanthocybium solandrei*) or *Scomberomorus* spp., which are pelagic fishes with elongate trunks.

*Amazichthys* has residuals of 0.158 in the all-taxon OOL model, similar to that of elongate or "semi-elongate" fishes like *Esox* spp., *Sphyraena* spp., *Istiompax indica*, and *Kajikia audax* (Supplementary File S3: Table S17).

Despite *Amazichthys*' elongate body shape, this taxon still has pelvic fins located immediately posterior to the end of the ventral armor, and the pelvic girdle is still located approximately midway between the pectoral fin and base of the caudal fin [21]. This supports the idea that these patterns are present across all arthrodires, even those with atypical body shapes. *Amazichthys* maintains the close relationship between the pelvic girdle and ventral armor seen across arthrodires despite its elongate body plan due to having a highly elongate ventral shield. This suggests that thoracic armor shape is indicative of body shape in arthrodires, and other arthrodires with *Amazichthys*-like body plans can be identified in the fossil record based on elongate ventral shields. Such a feature is not present in *Dunkleosteus*, where the ventral shield is unusually short compared to the rest of the armor. Thus, the morphology of *Amazichthys trinajsticae* is not suggestive of a more elongate shape and longer body for *Dunkleosteus terrelli*. Indeed, in some respects (e.g., in the correlation between armor shape or pelvic fin placement and body shape) the anatomy of *A. trinajsticae* actually supports the short, deep body plan for *D. terrelli* recovered here.

These conclusions on broader patterns of body shape evolution in fishes agree with the results of prior studies. Trinajstic et al. [57] note the dimensions of the abdominal cavity in arthrodires are more similar to that of osteichthyans, whereas chondrichthyans show an anteroposteriorly elongate body cavity. These patterns may be related to different buoyancy control mechanisms in different fish groups. Chondrichthyans have an anteroposteriorly elongate body cavity [57] because this allows an already existing organ (the liver) to be enlarged and used as the primary buoyancy organ without altering the arrangement of the viscera. This pattern is seen in almost all chondrichthyans with the exception of rays (Batoidea) and chimaeroids (crown Holocephali), including stem-holocephalians such as the symmoriforms *Cladoselache* and *Ferromirum* [204,205]. Osteichthyans, on the other hand, control buoyancy through the development of an entirely novel organ (the lungs/swim bladder). The swim bladder must compete for space in the body cavity with the existing viscera and musculoskeletal system, but also requires a connection to the oropharyngeal cavity (at least in physostomous fishes, which is the ancestral state [74]). In this case, the simplest functional solution would be to expand the body cavity dorsoventrally (as in most actinopterygians). This allows the body to be partitioned vertically and allow more room for organs in the dorsoventral axis, similar to adding stories to a building to increase space. In this regard, the functional significance of the short, deep body of arthrodires and other placoderms, which have neither a swim bladder nor a liver that is enlarged to the degree of chondrichthyans [57], is unclear. However, preliminary research by the present author has identified several biomechanical aspects of the arthrodire body plan that might select for such a body shape [115].

In conclusion, almost no living fish has a body shape like an arthrodire. Although there are many anteroposteriorly short and dorsoventrally deep (=discoid) teleosts, these taxa still retain a typical mediolaterally narrow actinopterygian body plan. Some of the only extant fishes that approach arthrodires in body shape are lungfishes and coelacanths (which have deep bodies but are also much wider mediolaterally than actinopterygians) and tunas (which are actinopterygians but tend towards having a wider body in cross-section). In terms of the ratio of body cross-sectional area to total length, *Dunkleosteus* is most similar to opah (*Lampris* spp.), true tunas (*Thunnus* spp.), and lamnids (e.g., *Carcharodon carcharias*), and to a lesser degree other large pelagic or semi-pelagic fishes such as billfishes (Istiophoriformes) and the more robust members of *Carcharhinus* (*C. leucas*, *C. obscurus*). Overall, while the present study offers new solutions to the long-standing issue of body size in arthrodires, it also expands the body shape diversity of fishes and raises new challenges by demonstrating arthrodires have body shapes that are non-analogous to the two major groups of extant gnathostomes. This, in turn, would be expected to have a significant

effect on aspects of the functional morphology and paleobiology of arthrodires, such as swimming kinematics.

*4.3. Body Size of Dunkleosteus terrelli*

Ever since the remains of giant arthrodires like *Dunkleosteus terrelli* were first discovered in the Devonian shales of eastern North America, people have wondered at the size of these fishes. Even in the original description of *D. terrelli* Newberry ([206]: p. 24) wrote: "I have been frequently asked by those examining the bones of *Dinichthys*, what was the probable size of this great fish?". Unfortunately, due to the mostly cartilaginous endoskeletons of arthrodires, the answer to this question has generally been "as large as you want them to be". This observation, again, goes back to Newberry ([207]: p. 315), who wrote: "we know that [*Dinichthys*] could not have been less than 2 $\frac{1}{2}$ to 3 feet in diameter, but it is impossible to say whether the fish was 10 or 15 feet in length". Because the present study produces estimates that are so different from typically cited lengths for *Dunkleosteus terrelli*, it was necessary to review these prior estimates to try and resolve this discrepancy. Extraordinary claims require extraordinary evidence, and so extraordinary evidence was required to show previous estimates of 5+ m for *D. terrelli* were not supported.

Reviewing prior length estimates for *D. terrelli* (Table 7), two major trends become apparent. First, as previously noted by Ferrón et al. [12], most prior size estimates of *Dunkleosteus terrelli* are speculative and not based on explicit quantitative methods. Length estimates for *Dunkleosteus* are often provided without explaining how these values were calculated or what measurements were used to produce them. These studies sometimes imply that the length of *D. terrelli* was estimated using proportions of smaller arthrodires like *Coccosteus*. This is the case for Newberry [206]'s estimate for *Dunkleosteus* and Dean [208]'s estimate of ~5 m for *Titanichthys clarki*. However, even in these cases it is not stated what elements of *Coccosteus* were used to estimate size. This is a concern because the plates of many arthrodires show allometric growth [22,209], and assuming isometry between arthrodire plates of different sizes may not be possible. Only two prior studies have estimated size in *Dunkleosteus* explicitly without scaling from *Coccosteus*. These are Hussakof [116], who estimated length based on "entering angle of the body" (see [150]), and Ferrón et al. [12], who estimated length using upper jaw perimeter in extant sharks. These are also the only studies in which the methods used to estimate length are clearly stated, as well as the only ones (along with Johanson et al. [29]) that refer to specific specimens of *Dunkleosteus* when estimating length. Thus, these are the only studies whose length estimates can be independently tested and potentially replicated.

**Table 7.** Previous length estimates of *Dunkleosteus terrelli* arranged in chronological order and their methodology. "Unstated" refers to estimates where the methodology used to calculate these length estimates is undefined and no citation is made to length estimates in prior studies.

| Study | Length Estimate | Method of Estimation |
|---|---|---|
| Newberry [206]: p. 24 | 4.5–5.5 m ("15 to 18 feet") | Extrapolated from *Coccosteus cuspidatus* |
| Newberry [210]: p. 24 | 4.5 m ("15 feet in length") | Unstated (implied correlation with *Coccosteus*) |
| Dean [211]: p. 130 | 3 m ("10 feet") | Unstated |
| Hussakof [116]: pp. 32–34 | 1.67 m (juvenile) [1] <br> 2.43 m ("8 feet", juvenile) [1] <br> 3.79 m (extrapolated CMNH 5768) | "Entering angle" of body (sensu Dean [150]). |
| Anonymous [212] | 7.6 m ("25 feet") | Unstated [2] |
| Hyde [213] | 4.5–6 m ("15 to 20 feet") | Unstated |
| Romer [214]: p. 49 | 9 m ("may have reached a length of 30 feet") | Unstated |
| Colbert [215]: p. 36 | 9 m ("30 feet") | Unstated |
| Denison [61]: p. 88 | 6 m | Unstated |
| Williams [216] [3] | 5 m | Unstated |

**Table 7.** *Cont.*

| Study | Length Estimate | Method of Estimation |
|---|---|---|
| Maisey [217]: pp. 80–81 | 4 m (figured specimen) 5–6 m (typical adult) | Unstated |
| Janvier [218]: p. 12 | 6–7 m | Unstated [4] |
| Young [7] | "6 m, with evidence that some individuals may have doubled that length" | Unstated |
| Anderson and Westneat [8] | 6 m | Unstated |
| Anderson and Westneat [9] | 10 m | Unstated |
| Carr [11] | 4.5–6 m | Unstated |
| Long [6]: pp. 88–90 | 4–8 m | Unstated [5] |
| Sallan and Galimberti [32] | 8 m | Stated to be from Denison [61], but cited length disagrees with latter study. |
| Ferrón et al. [12] | 6.88 m (CMNH 5768), 8.79 m (maximum) | Upper jaw perimeter |
| Long et al. [219]: p. 13 | 6–8 m | Unstated [4] |
| Johanson et al. [29] | ~3 m (juvenile) [6] ~7.1 m (extrapolated CMNH 5768) [6] | Unstated |
| Present Study | 3.4 m (typical adult = CMNH 5768), 3.9–4.1 m (maximum) | Orbit-opercular length |

[1] Estimate based on an individual of "*Dinichthys intermedius*" (=juvenile *D. terrelli*) in the AMNH (specimen number unknown) with an inferognathal 31 cm long and a skull roof 27 cm long. Adult length extrapolated assuming similar head–body proportions for CMNH 5768, length estimated based on entering angle in CMNH 5768 can be found in Table 3. [2] Publication date (1923) and context suggest that this is a field estimate referring to one of the mounted *Dunkleosteus* specimens at the CMNH or USNM. [3] Semi-popular account but treated as primary reference in Hansen [220], so considered here. [4] Mentions specimens with "carapaces" (head and thoracic armor) over 2 m long, significantly larger than any specimen in the collections of the AMNH, CMNH, or NHMUK (~75% of the hypodigm), but do not provide specimen numbers. [5] Mentions specimens "with headshields over a meter long", significantly larger than any specimen of *D. terrelli* in the collections of the AMNH, CMNH, or NHMUK (~75% of the hypodigm), but do not provide specimen numbers. It is possible this estimate is referring to Denison [61] or Janvier [218], but this is unclear. [6] Authors suggest length of ~3 m for studied specimen. Assuming similar head–body proportions this would produce length of 7.1 m for CMNH 5768.

However, even when estimating the length of *Dunkleosteus terrelli* using the proportions of *Coccosteus cuspidatus* it is not possible to replicate the larger length estimates considered typical for this species. For example, assuming *Dunkleosteus* exhibited similar head–body proportions as the reconstruction of *C. cuspidatus* in Miles and Westoll [15], CMNH 5768 is estimated as 341 cm long (Figure 11, Table 3). Using the mediodorsal plate, which is known to show significant inter- and intraspecific allometry in arthrodires [22,209,221] and thus could be responsible for higher estimates, results in a length of 223 cm using the midline length of the mediodorsal (as in [64]) or 297 cm using the entire external length of this plate (Table 3). The posteroventrolateral plate, which shows isometric growth in arthrodires and thus is considered a more reliable indicator of size ([209]; J. Long, pers. comm.), produces a length of 388 cm (Table 3). The length of the inferognathal produces a length of 523 cm for *Dunkleosteus* (Table 3), but coccosteomorphs have smaller, more subterminal mouths than dunkleosteoids [33,222,223]. Thus, the inferognathal length of coccosteomorphs would be expected to overestimate length in *Dunkleosteus*. Aside from mouth dimensions [12,33], one of the few measurements that does produce larger size estimates in *D. terrelli* is body depth, which produces an estimate of 614 cm for CMNH 5768 (Table 3). However, as noted above, *D. terrelli* has an unusually deep body among arthrodires and thus body depth is considered an unreliable estimator. Thus, even when estimating the length of *D. terrelli* using the dimensions of smaller arthrodires the resulting lengths are more similar to the results of this study than previous estimates.

Second, estimated lengths of *Dunkleosteus* have gotten larger over time. The earliest studies on *Dunkleosteus terrelli* [116,206,210] infer lengths of about 3–5 m, similar to, if slightly higher, the estimates produced here. However, starting in the 1920s, size estimates for *D. terrelli* greatly increase such that values of 7–9 m become more common. The reason for this increase in estimated length is unclear. It is tempting to attribute this to

a better understanding of size variation within *Dunkleosteus*, correlated with the large number of arthrodire fossils collected during the early 20th century Cleveland construction boom [117,212,213]. Indeed, both CMNH 5768 and CMNH 5936 were collected during this period. However, specimens of *Dunkleosteus* comparable in size to CMNH 5768 have been known since Newberry ([210]: plate 33, referring to AMNH FF 108). Instead, at least some of these larger estimates appear to be the result of lapsus, exaggerations, and overly large initial field estimates (e.g., [212,214,224]). Thus, much like a fish tale about 'the one that got away', *Dunkleosteus* became increasingly larger with each retelling, particularly without re-examination of the original material or more rigorous methods of estimating this taxon's body size. This issue was exacerbated by the fact that, due to a long series of historical accidents extending as far back as the late 19th century ([225], D. Chapman pers. comm., A. McGee pers. comm., E. Scott pers. comm), the fishes of the Cleveland Shale have been historically understudied compared to both Devonian vertebrates as a whole and relative to *Dunkleosteus* and *Cladoselache*'s importance in evolutionary history [49] and the paleontological "canon". The present situation with *Dunkleosteus* shows how easy it is to take common knowledge for granted, especially for poorly known or extinct organisms.

After the middle 20th century, where estimates of 9–10 m were common, lower estimates of 5–6 m for *Dunkleosteus terrelli* became common consensus (e.g., [11]). This was the case until Ferrón et al. [12], which estimated lengths of 6.9 m for large, adult individuals of *Dunkleosteus* (i.e., CMNH 5768), and as large as 8.79 m for the very largest specimens (CMNH 5936) based on upper jaw perimeter. However, as previously noted by Engelman [33], these length estimates are suspect, as arthrodires have much larger mouths than extant sharks, mouth dimensions fail to accurately estimate body length in smaller arthrodires, and length estimates using mouth dimensions require unusually short heads and hyper-anguilliform postcrania for most arthrodires (which are not present in whole-body arthrodire fossils). The reconstruction of *Dunkleosteus* presented here may also have unusual proportions (i.e., its relatively short, deep body), but in this case the reconstructed body shape agrees with other osteological dimensions of the specimens (i.e., their deep thoracic armor) and resembles proportions in other arthrodires.

The present study produces length estimates for *Dunkleosteus terrelli* that are significantly lower than almost all previous estimates for this species (compare Tables 3 and 4 with Table 7). Typical adults of *Dunkleosteus terrelli*, as represented by CMNH 5768, are estimated as 3.4–3.5 m in length, with the very largest individuals of this species reaching perhaps 3.9–4.1 m (with a maximum possible length of 4.95 m based on %PE). These estimates are based on variables that accurately predict body length across a broad sample of fishes (lampreys, chondrichthyans, sarcopterygians, and actinopterygians), including arthrodires known from complete remains (e.g., *Coccosteus*, *Watsonosteus*), suggesting these methods should accurately predict body length in *Dunkleosteus terrelli*. The 95% prediction intervals of these methods are wide, but this appears to be driven by statistical issues with calculating prediction intervals for log-detransformed data, with other method of calculating uncertainty (like %PE) suggesting lengths greater than 5 m are highly unlikely. These shorter length estimates for *D. terrelli* remain consistent (i.e., between 3.1 and 3.5 m) even under different model assumptions, like estimating size using only sharks, pelagic fishes, or fusiform fishes. It is possible these estimates could be inflated slightly, perhaps by 10-20%, based on the error rates seen in arthrodires and the overall %PE of the model, but this is not sufficient to produce lengths of 5+ meters which have traditionally been considered typical for *D. terrelli*.

It should also be noted that the higher length estimates for *Dunkleosteus* do not actually make the animal larger in overall size, merely longer relative to the dimensions of the armor. The dimensions of the head and armor in *Dunkleosteus* represent a hard constraint on any attempt to reconstruct this taxon, as it can only be so long relative to its armor before such lengths become improbable for a pelagic, fusiform fish [33]. This is particularly the case with regard to head length, the relative size of the gill chamber, the location of the pectoral and pelvic fins, and the fact that fossils of *Dunkleosteus* show a very deep body relative to its

length. In order for *Dunkleosteus* to reach lengths of 5+ meters, it would need to have body proportions that not only dramatically different from any living fish, but also not resemble any arthrodire for which complete remains are known. In fact, the head-body proportions in the present reconstruction of *Dunkleosteus* are already close to the limits of variation seen in fishes without elongate trunks (head is ~18% total length). Additionally, every other potential size proxy in *D. terrelli* (relative positions of pelvic and pectoral fins, "entering angle", scaling off of non-oral dimensions using other arthrodires, etc.; Supplementary File S3) would need to be spurious. There are currently no anatomical lines of evidence which uncritically support larger lengths for *D. terrelli*; measurements that do are unlikely due to being confounded by ecology (body depth) or phylogeny (mouth dimensions).

That said, although *Dunkleosteus* appears to be much shorter than previously thought, it also appears to be extremely massive relative to its length. When estimating the body mass of arthrodires using a multivariate ellipsoid equation, arthrodires are found to be much more massive relative to their length than sharks (Figure 10). This is largely due to the fact that arthrodires have shorter, deeper, and wider bodies than sharks when scaled to the same body mass (Figure 12). For example, CMNH 5768 is estimated to weigh approximately 1008.4 kg and measure 341 cm in length. In order for a typical (i.e., non-lamnid) shark to reach similar weights, it would need to add another 161 cm of length (i.e., measure 502 cm in total length; Supplementary File S3: Section S17.6). Notably, this is only considering the flesh mass of arthrodires, and thus arthrodires may have been even heavier due to their dermal armor. However, as noted under Results the contribution of the armor to the total weight of *Dunkleosteus* may be smaller than previously believed.

### 4.4. Body Size Evolution in Paleozoic Vertebrates

The revised size estimates for *Dunkleosteus* presented here have significant implications for our understanding of vertebrate evolution. The Devonian is considered a key period in vertebrate evolution, as this is thought to be when vertebrates first evolved large body size. Traditionally, vertebrates have been viewed as undergoing an explosive increase in body size during the Devonian. Prior to the Devonian, vertebrates are generally <1 m in length with most being <35 cm (but see Choo et al. [226]). However, by the end of the Devonian vertebrates are considered to show body size distributions comparable to modern marine faunas [32]. Several studies have proposed abiotic or biotic drivers for this phenomenon, including sudden increases in atmospheric oxygen [227] or competitive interactions with eurypterine eurypterids [228,229]. Some studies have even found evidence for a peak in vertebrate body size in the Famennian followed by a reduction in body size across the Hangenberg Event and Devonian–Carboniferous boundary [32], which is strongly influenced by body-size patterns within placoderms. Arthrodires like *Dunkleosteus* have traditionally been used to establish the upper limits of vertebrate body size throughout the Devonian, with the appearance of *Dunkleosteus* and other large arthrodires in the Frasnian–Famennian considered to represent the oldest occurrence of vertebrate megafauna [6–9].

However, with these revised estimates of body size in *Dunkleosteus*, there appears to be no strong evidence that vertebrates exceeded sizes of 5 m prior to the Carboniferous. Devonian vertebrates do not even appear to reach the sizes spanned by modern marine fishes, with the largest individuals of *Dunkleosteus terrelli* being significantly smaller than the largest known individuals of *Carcharodon carcharias* in both length and mass (Figure 13). Other latest Devonian arthrodires are either similar in size to *Dunkleosteus* or much smaller. The elephant in the room when discussing the maximum size of arthrodires is the aspinotho-racidan *Titanichthys* spp., which has been interpreted as a very large suspension-feeding planktonivore similar to a whale shark (*Rhincodon*) or basking shark (*Cetorhinus*) [230], and hence might be expected to reach larger sizes than *Dunkleosteus*. The largest specimen of *Titanichthys* the author could locate is AMNH FF 7134, the complete, flattened specimen currently on display in the AMNH Hall of Vertebrate Origins. This specimen is only slightly larger than CMNH 5768 [208], and OOL using the model with shape and variable slope for Chondrichthyes produces a length estimate of 4.15 m (+/−%PE: 3.64–4.66 m; see Online

Supplementary Information, Table S17). Using only data from large filter-feeding sharks (N = 24) produces smaller lengths of 3.34 m (+/−%PE: 3.12–3.56 m) for this individual (see Online Supplementary Information, Table S17). Admittedly, the range of body sizes spanned by *Titanichthys* is not as clearly defined as *Dunkleosteus* because remains of this taxon tend to be fragmentary [22], but *Titanichthys* appears to have been similar in size or only slightly larger than *Dunkleosteus*.

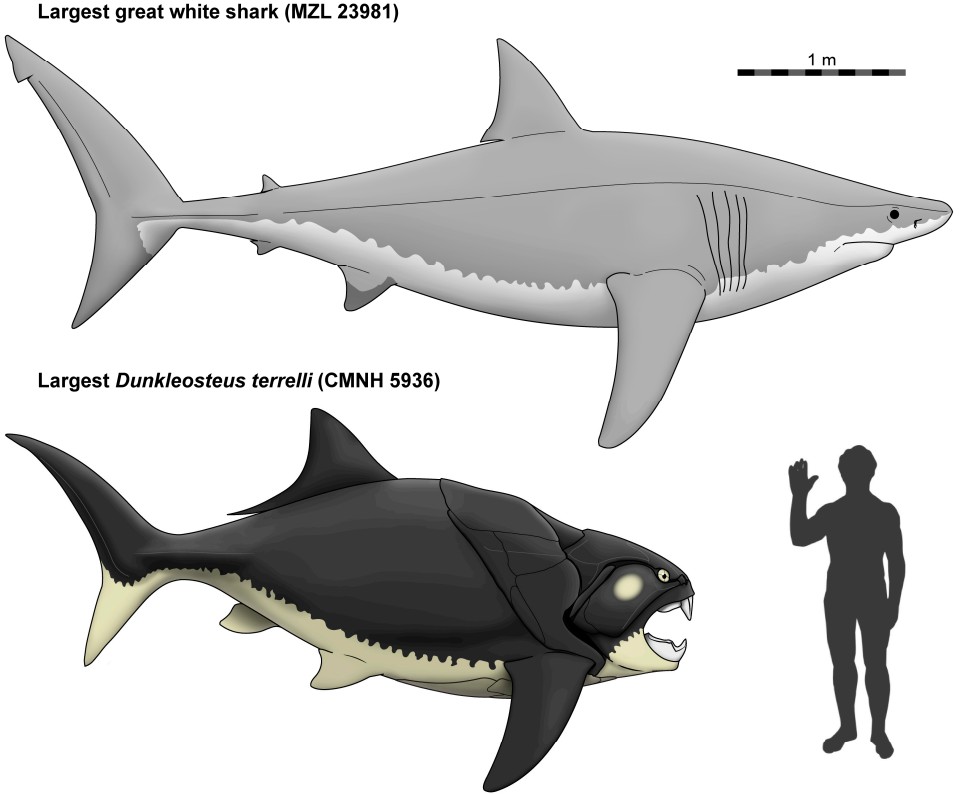

**Figure 13.** Reconstruction of the largest known specimen of *Dunkleosteus terrelli* (CMNH 5936, proportions primarily from the slightly smaller but more complete CMNH 5768), compared to one of the largest reliably measured specimens of *Carcharodon carcharias* (MZL 23981; [166]). A 175 cm tall human for scale (from NASA). Drawings by Russell Engelman.

Other groups of Devonian vertebrates appear to have reached similar maximal sizes as arthrodires. For example, while arthrodires have frequently been reconstructed as unstoppable apex predators of Devonian ecosystems, at least some Cleveland Shale chondrichthyans reached comparable sizes to *Dunkleosteus*. CMNH 5238 is an undescribed ctenacanth shark that preserves a Meckel's cartilage 68 cm in length [231]. Ginter [232] suggested CMNH 5238 pertained to an animal 5 m in length, though how this estimate was produced is not detailed. Assuming similar proportions to the ctenacanth *Dracopristis hoffmannorum* [233], this produces a crude estimated total length of ~4.2 m, implying an animal similar in length to the largest known individuals of *Dunkleosteus*. However, it is likely this chondrichthyan weighed much less given the more slender body proportions of chondrichthyans compared to arthrodires. Estimating the length of CMNH 5238 using the proportions of *Goodrichthys eskdalensis* (NMS 1950.38.46, see Maisey et al. [234]) also produces an estimated body length of 4.2 m. A more rigorous estimate of this specimen's body size is beyond the scope of the present study but under investigation by the author.

Outside of the Cleveland Shale, several species of tristichopterid sarcopterygians (*Edenopteron*, *Hyneria*, and *Eusthenodon*; see [235–237]) are considered to reach sizes of ~3 m. Unpublished specimens suggest that *Hyneria* could have grown even larger, possibly up to ~3.5 m in length (scaling from lower jaw length estimate in [235] and proportions of

*Eusthenopteron* in [135]). Similarly, an undescribed maxilla of *Onychodus* from the middle Devonian Delaware Limestone of Ohio [235] suggests onychodontids could reach lengths of ~4 m. Overall, these estimates indicate an overall congruence in maximum body size among Devonian vertebrates, given at least five different groups (dunkleosteoid and aspinothoracidan arthrodires, ctenacanth chondrichthyans, and onychodontiform and tristichopterid sarcopterygians) all seem to reach maximum lengths of about 3.5–4.2 m.

Thus, vertebrates may not have reached sizes ≥ 5 m, that is, comparable to or greater than modern marine megafauna like whales and large sharks [238,239], until the Carboniferous. One would expect the first vertebrates to have reached such sizes to be the rhizodonts, which have often been reported as reaching lengths of 6–7 m [240], but this may not be the case. Jeffery [241] estimates *Rhizodus hibberti* to have measured approximately 5.63 m in length by scaling from a juvenile individual of *Strepsodus anculonamensis* [242]. Estimating the total length of *R. hibberti* from *Goolongongia loomesi* [243], which is larger than *S. anculonamensis* but more distantly related and is dorsoventrally flattened, unlike Carboniferous rhizodonts (J. Jeffery, pers. comm.) produces a length of 5.14 m. This is longer than *Dunkleosteus*, but rhizodonts also have much more elongate bodies, and estimated masses for *Rhizodus* under an ellipsoid model are about 1000–1500 kg (Supplementary File S3: Table S17), similar to those produced for *Dunkleosteus terrelli* here. Thus, although *Rhizodus* appears to have grown slightly longer than *Dunkleosteus*, the two taxa were likely comparable in body mass. Another potential contender for the first vertebrate to reach sizes comparable to extant marine megafauna is the undescribed material of *Saivodus striatus* from the Mississippian (Visean) St. Louis/Ste. Genevieve Formations of Mammoth Cave, Kentucky, USA [244–246], estimated as potentially being 6–7 m in length (J.-P. Hodnett, pers. comm.), but this is heavily dependent on how the length of this animal is estimated.

The earliest vertebrates that are well-supported in reaching or exceeding sizes seen in modern marine megafauna (e.g., *Carcharodon*, *Rhincodon*, cetaceans) are the edestoid chondrichthyans of the late Carboniferous, of which the Moscovian (~310 Ma) *Edestus heinrichi* has been estimated as potentially measuring 6.7 m [168]. A reconstruction made by the author as a visual aid for comparison with *Dunkleosteus* [247] based on the skull dimensions of FMNH PF2204 [248] and general anatomical proportions and patterns of variation seen in eugeneodonts (e.g., [249]), other chondrichthyans (such as lamnids), and broader allometric relationships among fishes (e.g., OOL), results in a lower tooth whorl that almost perfectly matches the size of ANSP 22393, the largest known individual of *E. heinrichi* [168], when scaled up to 6.7 m. ANSP 22393 was not considered when producing this reconstruction, and the fact that the scaled-up reconstruction matches the dimensions of this specimen is more or less a happy accident. Although clearly not the most rigorous manner of evaluating body length estimates, the fact that the present author accidentally and independently produced a reconstruction that agrees with the proportions of ANSP 22393 supports the 6.7 m estimate proposed by Tapanila and Pruitt [168]. All of these observations potentially imply a much slower path to large body size in vertebrate evolution (compare with Choo et al. [226], who proposed similar conclusions when looking at late Silurian vertebrates), and a need to revise our understanding of patterns of body size evolution in early vertebrates.

Indeed, it is possible arthrodires may have even suppressed the evolution of large vertebrates prior to the Carboniferous. One key reason why animals evolve large body size is predation, either to become too large to be a viable target for predators [250,251] or to be able to capture larger, higher-caloric prey [252–254]. In arthrodires, which have unusually large mouths relative to body length [33], there would be reduced selective pressure for the evolution of larger body sizes. This is because, based on mouth size, a 3.5 m arthrodire can effectively fill the same ecological role as a 5.5 m shark [33]. Similarly, with the large mouths and cutting mouthparts of arthrodires, there would be reduced pressure for the evolution of larger body sizes as a defense against predation. Large size becomes a less effective defense mechanism when the potential predator has a disproportionately large mouth and can circumvent gape limitations by simply biting chunks out of prey. This

would allow arthrodires to attack comparatively larger prey relative to their body length than modern sharks or bony fishes [33,231,255]. This agrees with previous studies that have suggested the ecological "rules" for the Devonian were very different from the present day, driven in part by the presence of arthrodires as dominant predators [256,257].

Perhaps most importantly, these well-constrained body length estimates open the door for new and more rigorous studies of arthrodire paleobiology. Our understanding of the paleobiology of arthrodires and other placoderms has largely lagged behind all other vertebrates, and one of the biggest reasons for this is the lack of rigorous size estimates for these taxa. As mentioned above (see Introduction) body size is an important factor to consider even in analyses as simple as comparing the morphology of specimens of different sizes. This has forced every paleobiological study involving arthrodires to either tiptoe around this issue or use estimates based on other regions of the body that may not be a reliable proxy for size (or use approximate guesstimates, see Table 7). Many available size proxies are also based on arthrodire plates, and thus limit paleobiological comparisons between arthrodires and other vertebrates like chondrichthyans or osteichthyans. The present study provides a new, rigorous way to estimate body size in arthrodires, allowing for direct comparisons of proportions between arthrodires and extant fishes as well as more reliable size estimates for studies of Devonian paleoecology and broader patterns of vertebrate evolution.

## 5. Conclusions

Head proportions, specifically orbit-opercular length, are a reliable predictor of body length in fishes. Allometric relationships between head and body proportions and head and body fineness ratios are consistent across a broad diversity of fishes, including lampreys, arthrodires, chondrichthyans, actinopterygians, and sarcopterygians. As a general rule, short fishes have short heads, and long fishes have long heads. Head–body proportions are much more strongly constrained in non-tetrapods ("fishes") than tetrapods, given nearly all fishes follow a single allometric relationship (though sharks confound this due to their non-random distribution of body shapes/life habits with respect to size).

Typical adults of *Dunkleosteus terrelli*, traditionally considered to measure 5–8 m in total length, are estimated to measure ~3.3–3.5 m using head proportions (OOL). The very largest individuals of *D. terrelli* are estimated as measuring ~4.1 m. Almost all model permutations produce lengths of 3.1–3.5 m for typical adults of *D. terrelli*, represented by CMNH 5768 (the specimen that serves as the basis for the majority of *Dunkleosteus* casts seen around the world). This occurs despite these models starting from different starting assumptions and in some cases using different variables or samples of modern taxa. The only methods that return higher estimates (i.e., mouth dimensions and body depth) are likely to be spurious based on comparative anatomy. Some models (e.g., head length by itself) actually produce even smaller lengths than the best-fit estimates reported here, but are also unlikely on anatomical grounds.

The shorter lengths for *Dunkleosteus* estimated here better agree with the preserved anatomy of this taxon and other arthrodires. Arthrodires as a whole tend to be characterized by anteroposteriorly short, rotund body plans, distinctly unlike either elasmobranchs or actinopterygians. These results dramatically reduce the size of the largest known Devonian vertebrates, and suggests that multiple Devonian fish groups (arthrodires, chondrichthyans, and sarcopterygians) reached similar maximum sizes. Based on these results, there is no strong evidence for vertebrates greater than 5 m in length prior to the Carboniferous, and the mode and tempo for the "explosive" expansion in vertebrate size during the Devonian needs to be reassessed.

**Supplementary Materials:** The following supporting information can be downloaded at: https://www.mdpi.com/article/10.3390/d15030318/s1. Supplementary File S1. Database of measurements of extant fishes and arthrodires used to estimate the total length and shape of *Dunkleosteus terrelli*. Supplementary File S2. Measurements of specimens of *Dunkleosteus terrelli* collected in this study to estimate the size of the largest known specimen of *Dunkleosteus* (CMNH 5936). Supplementary File S3.

Knitted .html document showing supplementary statistical analysis and documentation for this study. [11,12,15,16,22,39,69,93,95,115,116,118,121,123–125,133–135,140,150,162,166,208,242,243,258–272]. Supplementary File S4. List of supplementary references used in Supplementary File S1. Supplementary File S5. Silhouettes of *Dunkleosteus terrelli* (CMNH 5768), showing the estimated length and possible range of variation (using ± PE) using the best fitting model (with shape and allowing different slopes between Chondrichthyes and all other fishes). Note how the length at the lower range of the interval creates an unreasonably short body and the length at the upper end is still significantly shorter than most prior estimates of *D. terrelli*. Supplementary File S6. Skeletons of arthrodires with post-thoracic remains, with arrow showing the location of the pelvic girdle (or possibly the claspers, which are associated with the pelvic girdle; see Trinajstic et al. [154]) and its association with the posterior end of the ventral armor. (**A**), *Coccosteus cuspidatus* (ROM VP52664, from collections.rom.on.ca); (**B**), *Watsonosteus fletti* (NMS G.1995.4.2, courtesy of M. J. Newman); (**C**), *Incisoscutum ritchei* (WAM 03.3.28, modified from Trinajstic et al. [273]); (**D**), *Amazichthys trinajsticae* (AA.MEM.DS.8, modified from Jobbins et al. [21]). Scale = 5 cm, no scale available for **C**. Supplementary File S7. Silhouette of a small juvenile individual of *Dunkleosteus terrelli*, modeled after CMNH 7424. This image is needed to rerun the R code in Supplementary File S3. Supplementary File S8. Silhouette of a late-stage juvenile of *Dunkleosteus terrelli*, modeled after CMNH 6090 and 7424. This image is needed to rerun the R code in Supplementary File S3. Supplementary File S9. Silhouette of an adult individual of *Dunkleosteus terrelli*, modeled after CMNH 5768. This image is needed to rerun the R code in Supplementary File S3.

**Funding:** This research received no external funding.

**Institutional Review Board Statement:** Not applicable.

**Data Availability Statement:** The data presented in this study are available in Supplementary Files S1–S4.

**Acknowledgments:** The author thanks A. McGee, C. Colleary, H. Majewski, and R. Muehlheim (CMNH), J. Maisey and A. Gishlick (AMNH), B. Simpson, L. Grande, and C. McGarrity (FMNH), E. Price (FSBC), J. Kerr (MNHM), E. Bernard (NHMUK), S. Walsh (NMS), and M. Daly and M. Kibbey (OSU) for access to specimens in their care; O. Glaizot (MZL) for providing additional measurements of MZL 23918 which were used to create Figure 13; J. Newman for photos of *Watsonosteus*; S. van Mesdag and J. Boyle for access to their data; C. Engelman for assistance in taking measurements of preserved fishes; Y. Haridy for assistance in getting measurements from FMNH specimens, D. Croft, S. Simpson, M. Benard, P. Princehouse, R. Drushel, N. Gardner, R. Shell, R. Hawley, L. Bernstein-Kurtycz, R. Oldfield, J. Lundberg, J. Hannibal, M. Mihalitsis, J. Tait, G. Storrs, L. Weaver, R.B. Dahl, Y. Haridy, Z. Johanson, J. Long, K. Trinajstic, and M. Jobbins for helpful feedback; E. Scott and D. Chapman for discussions regarding the history of the Cleveland Shale; Y. Song for permitting the use of their silhouette of *Dunkleosteus* in the graphical abstract, R. Atit and Z. Johanson for discussions involving developmental biology in fishes; J. Jeffery, J.-P. Hodnett, and L. Tapanila for discussions regarding body size in Carboniferous vertebrates; F. Concha, A. De Santis, C. Duffy, R. Freitas, F. Mollen (Elasmobranch Research, Belgium), and R. Winterbottom for permission to use images of fishes they had collected, T. McCoy and R. Jones for being willing to provide data (even if that data could not be used in the final study), J. E. Randall for uploading his entire collection of fish images to be freely used on FishBase, and the four reviewers whose constructive criticism improved this manuscript.

**Conflicts of Interest:** The authors declare no conflict of interest.

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
