# Peer review of "A Devonian Fish Tale: A New Method of Body Length Estimation Suggests Much Smaller Sizes for Dunkleosteus terrelli (Placodermi: Arthrodira)"

_diversity, doi:10.3390/d15030318_

Round 1

Reviewer 1 Report

I have minor comments on the text that could be addressed, primarily with the respect to using lampreys as part of a phylogenetic bracket for arthrodires, particularly when there are a range of fossil jawless vertebrates that could also be considered. Also I've made some comments about the developmental observations. 

Author Response

Note: any changes requested by this reviewer not directly addressed in these comments (e.g., minor formatting changes, fixing typos) have been made without further comment.

“I have minor comments on the text that could be addressed, primarily with the respect to using lampreys as part of a phylogenetic bracket for arthrodires, particularly when there are a range of fossil jawless vertebrates that could also be considered.” (Comments to editor)

“In terms of a phylogenetic bracket, why not use fossil jawless fish, several groups of which are well-preserved?” (Line 153, page 5)

“I'm not sure what you mean by 'sufficient' here- lampreys are well removed from placoderms phylogenetically, but data from them wouldn't reflect data you would recover from fossil jawless fish?” (Line 178, page 5)

“And lampreys are derived in several features, would they provide this accurate prediction for arthrodires?” (Line 179, page 5)

Again, I think unless you include the jawless fossil vertebrates, you can't make these sweeping statements.” (line 1446, page 34)

These comments are addressed here because they all involve the issue of using lampreys in the present study. The primary reason I included lampreys as the outgroup were as follows…

  1. Lampreys are extant, which means their body proportions can be measured on complete, living specimens rather than extrapolated from fossil species. It also means their anatomy can be identified with little room for misinterpretation. For example, OOL is difficult to measure in many cephalaspidomorphs or galeaspids, because the gill openings of these animals are on the ventral surface but the orbits are dorsal and many specimens are preserved flattened in one of the two views. In other cases it is difficult to determine where the gill chamber ends and the posterior alimentary tract begins.
  2. I did not have access to decent samples of Paleozoic jawless fish fossils. The complete fish fossils I had access to tended to be arthrodires and a few Paleozoic chondrichthyans. I did try searching the literature for useable specimens of Paleozoic jawless fishes but few of these taxa appear to be known from complete remains [1].

I agree with the reviewer that extending these methods to test whether OOL accurately predicts length in jawless Paleozoic fishes is a logical next step for this study. However, the argument Paleozoic jawless fishes must be considered to allow broader applicability could also be applied to a large number of other now-extinct fossil fish groups: ostracoderms, antiarchs, rhenanidans, ptyctodonts, “acanthodians”, Paleozoic and Mesozoic actinopterygians, sarcopterygians, and chondrichthyans. There are limits to how extensive any study can be. OOL is shown to reliably predict body size in three extant gnathostome clades (Actinopterygii, Sarcopterygii, and Chondrichthyes), one clade of “Placoderms” (Arthrodira), and one group of very basal jawless fishes (Petromyzontiformes). At this point it seems safe to consider the phenomenon widely distributed in vertebrates, even if every single group is not tested here. The text has been amended to add that additional data from Paleozoic jawless fishes are needed to better examine how broadly distributed this pattern might be.

Additionally, much like the Methods section discusses with antiarchs and rhenanidans, many extinct Paleozoic jawless fishes have very unusual morphologies, and the assumptions OOL makes to estimate length based on nektonic, fusiform fishes may not be applicable to these other groups due to their ecology. Specifically, many Paleozoic jawless fishes are filter-feeders and/or have posteriorly located orbits, which produce bias when seen in living gnathostomes. Thus, phylogeny and ecology are confounded when trying to test if these taxa follow the pattern seen in other vertebrates.

This should not affect the overall results of the study. The primary concern here is using lampreys as the outgroup for phylogenetic bracketing. However, using complete arthrodires like coccosteomorphs alone would sufficiently bracket Dunkleosteus terrelli because it would result in this taxon fully bracketed by basal arthrodires, coccosteomorphs, and Amazichthys (see Figure 3). Lampreys were included in this analysis as an additional safeguard, since arthrodires are positioned basal to the rest of Gnathostomata.

“Both free-swimming? What are the differences in habitats?” (Line 76, Page 2)

This has been clarified in the revised text.

“Can you explain this in more detail, what are the bounds of the dataset- smallest size versus largest?” (Line 78, Page 2)

This is explained on the next line of the manuscript.

“But does this include the snout- longer snout (as part of the head) correlated with longer body which you aren't measuring?” (Line 138, Page 4)

I think both are true. OOL is pretty consistent but snout length also appears to be influenced by body elongation. For this reason I examined both head length and OOL when I compared fineness ratios of the head and body in Figure 3.1 in the Supplementary File 3. There is probably more going on here but because arthrodires universally have a smaller-than-expected snout compared to eugnathostomes trying to account for variation in snout length beyond that was beyond the scope of the present study’s focus on Dunkleosteus and arthrodires.

“Can you discuss your rationale for choosing the extant chondrichthyans and osteichthyans that you did?” (Line 199, page 6)

This is stated on lines 286-298 of the original manuscript.

“But you do include catfish?” (Line 253, page 7)

Antiarchs show a very different body plan from catfishes. Both are benthic animals, but antiarchs show a very unusual box-like body plan with jointed forelimbs like arthropod legs. Their bodies resemble no species of living fish (even the armored catfishes), and hence including them may not produce reliable results.

“why?” (Line 271, page 7)

“why? I think you need to expand upon this.” (Line 274, page 7)

These two comments are answered together here. The text has been updated to better support this conclusion. Camuropiscids have been considered active nektonic pursuit predators [2,3], but the remaining Gogo eubrachythoracids are somewhere on the spectrum between demersal and neritic in habits. Demersal is the null hypothesis due to being associated with a paleo-reef [2], but the swimming abilities of these taxa have never been studied and some may prove to be more active swimmers. Nevertheless, these taxa have to be categorized for the present study, and treating them as demersal requires the fewest number of assumptions at the present time.

“Phylogenetically? In terms of the phylogenetic bracket, would it be better to target basal taxa within these major clades?” (Lines 286-287, page 8)

Yes. The text has been updated to reflect this.

“not in holocephalans” (Line 314, page 8)

This has been corrected per reviewer comment.

“Also the postbranchial lamina is here, marking the posterior wall of the gill chamber” (Line 320, page 8)

This has been added per reviewer comment.

In actinopts, not sarcopts.” (Line 1480, p. 34)

There are two whole genome-duplication events prior to the origin of Gnathostomata (resulting in the four HOX clusters in non-actinopterygian gnathostomes, [4]), in addition to the ones within Actinopterygii. However, the manuscript agrees with what the reviewer is saying here: there are no whole genome duplications in sarcopterygians relative to actinopterygians that could explain the phenomenon under study. The text has been reworded to try and make this less ambiguous.

“Yes, I think you criticized it yourself, above.” (line 489, page 11)

This is true. However, I wanted to give proper credit to Ferrón et al. [5] as they stated this before me and to demonstrate I am not the only researcher to say this.

“What figure number is the graph above?” (line 655, page 16)

This appears to be a typo due to a glitch in how figure references can be formatted in Word. The effect is gone in the version I downloaded from SUSY.

“What part of the OOL does this affect?” (line 752, page 18)

The mandibular adductor originates from behind the orbit on the hyomandibula and/or preopercle. Thus, if this region is hypertrophied in Pygocentrus and related forms, it means that OOL is disproportionately long relative to body size because the taxon has to accommodate extremely large jaw muscles.

“the OOL?” (line 792, page 19)

Head length defined either as total head length or OOL, see Section 3 of the supplementary information.

“But researchers like Couly have found both somitic contribution and cranial mesoderm to the occiptial region in birds. In other papers it is the whole occipital region that is derived from cranial mesoderm, but in some papers it seems like this was determined from cranial bones not derived from neural crest.” (line 1418, page 33)

“What I took from this paper is that the segmentation pattern being compared is mediolateral, rather than rostrocaudal as in the somite clock. Would it be affected by the somite clock, has anyone tested this?” (line 1422, page 33)

“Are you saying that the head skeleton is controlled by the somitic clock? Neural crest makes a contribution to the anterior skull (including the orbital region), and see my comments earlier about what kind of segmentation is being compared in Amphioxus and the cranial mesoderm. Has anyone tested what you are proposing?” (line 1424, page 33)

I have rewritten this section based on some of these comments and further requests for clarification from the author. Specifically the idea this pattern might be controlled by the somitic clock may be incorrect. It is perhaps better attributed to anteroposteriorly-oriented morphogen gradients, which is the broader pattern of developmental control mechanisms that includes the somitic clock. However, the somitic clock is a specific example of this phenomenon with its own unique mechanisms, and thus may not be the precise example responsible for this pattern.

In terms of “Are you saying that the head skeleton is controlled by the somitic clock?”, this is partially the case. It looks like whatever is controlling the location of the head-trunk boundary in fishes is also correlated with axial elongation and anteroposterior axis patterning, such that the proportions of one body region mirror the proportions of the other. The anteriormost regions of the embryo are thought to be less subject to these kinds of controls (e.g., the cranial mesoderm is unaffected by the somitic clock), but this suggests broader morphogen patterns affect the head and trunk.

Additionally, in terms of “has anyone tested this?”, having talked with the reviewer I am not sure if anyone has. The reviewer has a stronger developmental biology background than I do but they were unaware of any literature that discusses the mechanism controlling the location of the head-trunk boundary in vertebrates, especially fishes. I had not found any similar studies in my survey of the literature. I had assumed whatever controlled this boundary was common knowledge among developmental biologists and so it was not discussed in the literature, but the evo-devo literature [6,7] suggests it has not been heavily studied.

             “What does this mean?” (line 1429, page 33)

Random walk is a commonly used term in evolutionary biology that describes the cumulative path of a random process over a period of time [8,9]. It’s often used to describe the evolution of quantitative traits within a lineage via the accumulation of mutations.

“endotherms?” (line 1721, page 41)

I think this is likely what is going on, and endothermy in Dunkleosteus has been suggested by previous authors [10]. However, going further with this argument would be speculative in the context of the present study.

“Some Gogo arthrodires have perichondral endoskeletons” (line 1737, page 41)

Yes, I think either Savanna van Mesdag or Kate Trinajstic said all arthrodires technically have perichondral endoskeletons, it is just bone is restricted to a very thin outer layer that is easily destroyed. The statement here has been modified.

“I think this came just from John Long- did I also say this?” (line 1792, page 43)

The reviewer is correct, this has been fixed.

“? Google tells me "The Latin term lapsi, 'lapsed', refers to Christians who, in contrast to the 'steadfast' (stantes) and the martyrs (martyres), renounced their faith during the persecutions, esp. in the 3rd cent."”

Fixed as per reviewer comment. The plural of lapsus is lapsus.

Reviewer 2 Report

This was a very exciting and transformational manuscript to review because the methodology was so compelling and the results so clearly obvious. Use of the OOL to determine body length is such a powerful tool, that determining body length in fishes will never be the same! This is a very important addition to the literature and will become the standard go-to reference for comparable works.

The figures and graphs were expertly composed/compiled.

I only had a few minor editorial comments/suggested changes.

Line 29: Change “D. terrelli” to “Dunkleosteus terrelli”

Line 75: Change “as not only is Dunkleosteus is likely” to “as not only is Dunkleosteus likely” remove the second “is”

Line 169: Change “the few arthrodires taxa” to “the few arthrodire taxa” remove the “s” from arthrodires

Line 286: Change “through an effort” to “though an effort”

Line 439: is this the best word order? “do yet not”

Line 472: Change “how well-the model” to “how well the model”

I look forward to seeing this work published.

Author Response

All corrections suggested by this author have been made without comment.

Reviewer 3 Report

This long manuscript entitled A Devonian Fish Tale proposes to revise the currently understood maximum length of the Devonian keystone predator Dunkleosteus. Using an array of statistical techniques, the author claims to have reduced the maximum length estimate of Dunkleosteus from 5+ meters to no more than 4 meters maximum length. The author then discusses the various implications of this finding, most notably the contention that maximum top aquatic carnivore megafaunal size, 6-7 meters body length, is not attained until the Late Carboniferous with the edestoid chondrichthyans at 310 Ma. This is an interesting claim, which the author further supports by arguing that dunkleosteoids did not need to max out in the superpredator size class because of their guillotine-like jaws, which could handle larger prey by biting them to bits. I have no cause to quarrel with the author’s statistical analyses. My primary problem with this manuscript is the author’s reconstruction of Dunkleosteus. The body proportions seem to be unrealistically compressiform, and contrast markedly with the profiles of Coccosteus and Amazichthys (with its elongate body). I think that a stronger case needs to be made in the revised manuscript that the paleoart presented here is what Dunkleosteus actually looked like. To remedy this, I recommend running the OOL analytical tools on the aspinothoracidid placoderm Dinichthys herzeri. What length and shape do the models return for D. herzeri? Does D. herzeri end up more closely resembling Dunkleosteus, or otherwise? I think that this would be a good test/control of the models developed here for estimating fish total body length from incomplete remains. And even with the new addition, shortening the text would considerably improve this manuscript.

Comments on the manuscript:

Line 109, Figure 1 caption, omit ‘is’

Line 239, fix precaudal typo

Line 1407 Schadian analysis might be more appropriate here than clock & wavefront mechanism

Line 1593 represent needs an s

Line 1984 give St. Lous back his i

Author Response

Responses to Reviewer 3

All minor comments (e.g., fixing typos) have been made without further comment. The only exception is the one regarding Schadian analysis: I have had difficulty finding out what Schadian analysis is and it is unclear how it can be applied to the present study. I understand Schadian analysis seems to advocate for a more holistic relationship between the proportions of various regions of the body. However, from what information I can find the principles of Schadian theories are vague and do not make much biological sense. Additionally, Schad's work seems to primarily focus on a threefold division between the head, visceral cavity, and limbs in tetrapods that is difficult to apply to fishes. I am not sure if Schad's ideas are well-supported in evolutionary biology, and am not sure if such an analysis would be appropriate here.

“The body proportions seem to be unrealistically compressiform, and contrast markedly with the profiles of Coccosteus and Amazichthys (with its elongate body). I think that a stronger case needs to be made in the revised manuscript that the paleoart presented here is what Dunkleosteus actually looked like.” (comments to editor)

I agree with the reviewer that the reconstruction of Dunkleosteus presented here appears unusually short and deep. However, this body shape is derived from the actual dimensions of the fossils. As mentioned in the Methods (see lines 527-529 of original manuscript), these drawings were made using 3D models and direct observations of mounted specimens of Dunkleosteus terrelli. Most specimens of Dunkleosteus have unusually deep thoracic armors for arthrodires; all of the complete subadult to adult individuals of D. terrelli at the Cleveland Museum of Natural History (CMNH 6090, 7054, and 5768) have thoraxes that are significantly deeper than their heads are long. This strongly constrains how slender-bodied a reconstruction of Dunkleosteus can be at any length. Even attempting to create an alternate reconstruction with a more slender thoracic armor suggests body depth can only be reduced by about 10 cm before the reconstruction conflicts with the proportions of the fossils.

This is part of the purpose of Figure 11 in the manuscript. If Dunkleosteus is scaled to match the proportions of other arthrodires, (e.g., keeping head length constant), this results in the proportions of other regions of Dunkleosteus, such as the location of the pelvic and pectoral fins, the association of the trunk armor with the thorax, and the close association between the ventral shield and pectoral girdle more closely agreeing with one another. However, this still results in Dunkleosteus having a very deep body compared to other arthrodires because of the pre-existing bony osteology of the thoracic armor.

This is actually a good example of why the present study is important. Dunkleosteus has a very, very unusual body armor shape compared to other arthrodires (even other dunkleosteoids), but this has been overlooked for many years because prior studies have uncritically applied a generic shark or eel-like body plan on the animal rather than looking at the dimensions of the fossils. The hypodigm of Dunkleosteus at the Cleveland Museum of Natural History has been largely unstudied from a morphological perspective (and this is occasionally noted in the literature, including by Heintz [1]: p. 115-116).

Some evidence for this shorter body plan in Dunkleosteus is already provided in the manuscript (see lines 1521–1601, pages 35–38 of the original manuscript). Some of the additional observations mentioned here (i.e., mounted specimens of Dunkleosteus invariably show a very deep body armor relative to the fossils' anteroposterior length) has also been added to the manuscript.

“To remedy this, I recommend running the OOL analytical tools on the aspinothoracidid placoderm Dinichthys herzeri. What length and shape do the models return for D. herzeri? Does D. herzeri end up more closely resembling Dunkleosteus, or otherwise?” (comments to editor)

As much as I would like to, this is not possible. No complete skull roof (or complete thoracic armor) is known for Dinichthys herzeri [2]. Complete thoracic armors are also unknown for the large arthrodires Titanichthys spp. and Gorgonichthys clarki. Some evidence suggests both Gorgonichthys and Titanichthys show body shapes similar to Dunkleosteus. Hussakof and Kepler [3] refer comparatively short ventral shield plates to Gorgonichthys, suggesting a body shape similar to Dunkleosteus. The immature Titanichthys reconstructed in Boyle and Ryan [4] (whose length is estimated in this study, see table 17.7 in supplementary info) shows a body shape more similar to Dunkleosteus than other arthrodires (though this specimen lacks a ventral shield and is a juvenile; specimens of Dunkleosteus show an ontogenetic increase in fineness ratio). However, none of the other large arthrodires are known in enough detail to use as a reliable test of body shape. Additionally, Dinichthys and these other large species are only distantly related to Dunkleosteus (they are aspinothoracidans, not dunkleosteoids). Eastmanosteus, which is a dunkleosteoid, shows a deep body relative to other arthrodires but not as extreme as in Dunkleosteus (which is to be expected given Eastmanosteus is geologically older and not specialized for pelagic life).

“And even with the new addition, shortening the text would considerably improve this manuscript.” (comments to editor)

I have tried to go through and make the manuscript less wordy where possible. 

  1. Heintz, A. The structure of Dinichthys: a contribution to our knowledge of the Arthrodira. In The Bashford Dean Memorial Volume, Gudger, E.W., Ed.; American Museum of Natural History: New York, 1932; pp. 115–224.
  2. Carr, R.K.; Hlavin, W.J. Two new species of Dunkleosteus Lehman, 1956, from the Ohio Shale Formation (USA, Famennian) and the Kettle Point Formation (Canada, Upper Devonian), and a cladistic analysis of the Eubrachythoraci (Placodermi, Arthrodira). Zool J Linn Soc 2010, 159, 195–222, doi:10.1111/j.1096-3642.2009.00578.x.
  3. Hussakof, L.; Kepler, W. On the structure of two imperfectly known dinichthyids. Bull Am Mus Nat Hist 1905, 21, 409-414.
  4. Boyle, J.; Ryan, M.J. New information on Titanichthys (Placodermi, Arthrodira) from the Cleveland Shale (Upper Devonian) of Ohio, USA. J Paleontol 2017, 91, 318–336, doi:10.1017/jpa.2016.136.

Reviewer 4 Report

Although I cannot follow hundred percent the paragraphs dealing with "deep statistics" and molecular biology, I consider the submitted paper to be original, novel and very well crafted under many aspects. I have only minor revisions to propose, all of which are detailed in the attached PDF.

Author Response

Note: any changes requested by this reviewer and not directly addressed in these comments (e.g., minor formatting changes, fixing typos) have been made without further comment.

“Elongate rostra do also contribute to make heads look elongate! Please consider applying some rephrasing here.” (line 137, page 4)

This line discusses head proportions in the general sense (i.e., including rostra). As mentioned later on in the study rostral length probably has some effect, but the purpose of this statement here is to mention how body proportions and head proportions (in the broad sense) are correlated in fishes.

“Maybe they could still be usable for predicting the precaudal length?” (Line 236, page 7)

I agree with what the reviewer is saying here. However, only one Gogo eubrachythoracid with near-complete post-thoracic remains has been described in the literature: Incisoscutum ritchei (see [1]), which is included in the present study. Other Gogo arthrodire specimens preserve parts of the spinal column but these fossils have not been published. I tried reaching out to the researchers working on the Gogo Formation eubrachythoracids but could not get any data on these undescribed specimens.

What about sunfishes, sturgeons and oarfishes...?” (Line 417, page 10)

This line is just an example of the kinds of fishes that were focuses on (i.e., fish megafauna). This statement has been clarified per reviewer suggestion.

“This sentence is a bit awkward and may be simplified.” (line 1026, page 24)

This has been done.

“I wonder whether lampreys are a good analogue here, given that they are not even gnathostomes.” (Line 1251, page 29)

In this context lampreys are being discussed because they have a larval stage, and this paragraph is discussing different allometric patterns in OOL between taxa with larval stages (lampreys and actinopterygians) and those without (sarcopterygians and chondrichthyans).

“You may mention, however, significant exceptions in many clades of terrestrial vertebrates.” (line 1425, page 33)

This is discussed further below, see lines 1473-1490 of the original manuscript. The text does present the caveat of this “often” occurring, but not guaranteed (as in the case of tetrapods).

“I wonder whether the last two sentence of this paragraph sound as an overinterpretation of the data.” (line 1472)

This section has been revised to clarify why it is thought the present data suggests “runaway selection” in certain species. Specifically, the outliers are more extreme than would be expected under a normal bell curve, suggesting that those outliers are experiencing very strong selective pressure for extreme morphologies. If they did not then the outliers in head body proportions would be expected to be much less extreme, and the distribution be normal.

“The reasons for that are actually unclear to me. Could you please be more specific?” (line 1690, page 40)

The text has been revised to present the evidence for this conclusion more clearly.

Amazichthys supports the short, deep body shape of Dunkleosteus because despite the very elongate torso of this taxon, the pelvic fins are still closely associated with the end of the ventral armor. This is associated with an extremely elongate ventral shield relative to the head and thoracic shield. This suggests two things:

1) The pelvic girdle is consistently associated with the end of the ventral shield in arthrodires, and because the pelvic fins are consistently located at about halfway along the length of the entire animal in non-acanthopterygian fishes suggests the same is true with Dunkleosteus. The position of the pelvic girdle relative to the tall armor suggests the body was deep in life, as supported by length estimates using OOL.

2) The elongate ventral shield of Amazichthys suggests armor proportions and trunk proportions are correlated in arthrodires. Specifically, the old models of Dunkleosteus where the thoracic armor is restricted to the very anterior end of the animal and most of the belly is exposed between the ventral shield and pelvic girdle (as in most shark-like reconstructions of D. terrelli) are unlikely. Dunkleosteus exhibits a very deep armor with an unusually short ventral shield compared to other arthrodires, suggesting it has a very short, deep body.

“Could you please disclose more of your finds on this issue?” (line 1714, page 41)

This statement has been clarified. The evidence for this conclusion was stated earlier in the paper (see lines 1614-1654 of the original manuscript, as well as section 7.5 in the supplementary information), this line is just meant to sum up the preceeding paragraph in a straightforward concluding statement.

This section has also been revised following this comment to make the association between this statement and the evidence in the preceeding paragraphs clearer. Arthrodires have a unique body shape in being very short yet dorsoventrally deep and wide for their weight.

“What about estimating the total length of this Titanichthys specimen accounting for the idisyncracies of magechasmids and cetorhinids (l.756 above)?”

This has been added.

“How these figures have been produced? I suspect based on the perimeter of the jaws, but please be more specific.” (Line 2013, p. 47).

The citation for this has been added, along with additional clarification as to how these estimates were produced.

  1. Dennis, K.; Miles, R.S. A pachyosteomorph arthrodire from Gogo, Western Australia. Zool J Linn Soc 1981, 73, 213-258, doi:10.1111/j.1096-3642.1981.tb01594.x.